# Fuller, Dworkin, Scientism, and Liberty: The Dichotomy between Continental and Common Law Traditions and Their Consequences

Nadia Elizabeth Nedzel

Southern University Law Center, Baton Rouge, LA 70813, USA; nnedzel@sulc.edu

**Abstract:** Dworkin's and other analytic/positivist philosophers' theoretical approach to law leads inexorably to politicization, totalitarianism, less justice, less trust in government, and less truth. A more practical approach is Fuller's, which is based on experience of human behavior and an analysis of what has worked in the past. That is also the approach traditionally used in the common law system. This article uses a comparative study of the two Western traditions, their history, and their most prominent legal philosophers to explicate how and why Dworkin's and Fuller's approaches are consistent and inconsistent with those traditions, followed by a comparative analysis of the results obtained by prominent international NGOs. Dworkin's approach, which grows out of analytic philosophy, is unworkable because like all scientistic theories, it treats human beings mechanistically, de-emphasizing personal responsibility, ignoring the need for individual incentive, and it assumes an all-encompassing, all-powerful government of experts to make legal decisions for a collectivity. Under Fuller's common law approach, the proper role of law is to manage conflict, as it cannot be prevented and cannot always be resolved, thus building the public's trust in government as unbiased and apolitical as possible. This concept of the rule of law places law above government, minimizes politicization, incentivizes personal responsibility, individual incentive, and entrepreneurship, and is the only true common good among men.

**Keywords:** rule of law; *rechtsstaat*; positivism; politicization; custom; common good; Dworkin; Fuller; justice; analytic philosophy

## 1. Introduction

Recently I visited the top of France's Cap d'Antibes, enjoying the expansive view from Italy to Cannes, and struck up a conversation with three other people: a local French couple and a Spanish tourist. Within five minutes we agreed that government in both Europe and the U.S. is overgrown, inept, intrusive, and often corrupt. If even the 'man (or woman) on the street' is concerned, the issue, then, among most lawyers and philosophers of law must be how to improve law and government. The Chair of the World Justice Project, William Hubbard, has indicated that the rule of law has deteriorated in 74% of countries worldwide (Hubbard 2022). Professor Senn has identified a societal lack of truth-telling and governmental failure to stop such falsehoods and provide justice as a fundamental problem—so where to begin to improve law, government, and humanity?

The controversial argument pressed by this article has three components. First, there is no final answer, no perfect approach, and one cannot study law without also considering its cultural context. Because law and government are social constructs and social institutions created by man, and are as mutable, fallible, and varied as man's habits, they cannot be accurately explained by science or the scientific method, and they do not share any underlying, hidden structure as posited by Dworkin or others in analytic philosophy or its antecedent positivist tradition. Second, modern western legal philosophy has two diverging views, that of Dworkin and the Continentalists and that of the common law tradition as exemplified by Lon Fuller's work. The nature of law (as with all human

institutions) is more accurately studied inductively, through history, custom, and culture than through efforts to develop overarching theories and hidden structures. Consequently, the third part of the argument is that as those concerned with an accurate and practical description of the nature of law, we should step away from that scientistic, positivistic view and examine what can be learned inductively from both traditions.

Lawyers tend to think that all such problems can be solved by law just as a man with a hammer sees everything as a nail. However, as a practical matter, we must recognize the limitations of law: it cannot force men to do good nor can it stop them from doing bad. It can only deter, punish, or reward. As James Madison profoundly quipped,

> "If men were angels, no government would be necessary. If angels were to govern men, neither eternal nor internal controls on government would be necessary."
> In framing a government which is to be administered by men over men, the great difficulty lies in this: you must first enable the government to control the governed; and the next place, oblige it to control itself."[1]

While Madison was referring to tyrannical acts by governments, in the modern age, continually expanding governments and ever-increasing legislation have decreased respect for both law and government. Alliances between big tech/business and big government (known as crony capitalism) have exacerbated the problem. Enlarging law or government to improve society is not likely to either improve society or create more justice. Experience proves that the opposite is the case: all-encompassing government encourages rent-seeking, falsification, and thwarts individual creativity and entrepreneurship. An assumption that government's role is to protect peace and not promote any agenda (i.e., that government should be a civil association and thus law should be non-instrumental) helps limit the politicization of government, thus encouraging truth and justice.

## 2. Material and Methods

This article uses research and inductive insight into the works of a wide number of scholars and historians over time. Inductive reasoning/explication is an alternative to deduction from first principles: one can use research into past practice to determine habits, customs, and principles that carry forward, and indeed this remains a major part of the legal analysis of common law—as Hume indicated, philosophy and history are deeply related (Livingston 1985, pp. 5, 22, 247–51; Hume 1756, p. 30). One can trace through history themes and cultural insights present in the legal institutions of both common law and civil law, going back to the beginning of recorded history. Moreover, those themes have remained the same over centuries: there is no rupture between the past and the present—the past life of institutions shapes their current character. "*[C]ar il n'existe pas de rupture entre le passé et le present: la vie passée des institutions a façonné leurs characters actuels.*" (Thireau 2009, p. 11) ("Because there is no break between the past and the present: the past life of institutions shapes their modern character."(trans. author).This type of research can be described as explication (Capaldi 1987, pp. 233–48), and is the methodology used by Lon Fuller, Friedrich Hayek, and Bruno Leoni in modern times, and it is the approach adopted here.

In contrast, exploration is an attempt to follow the implications of a hypothetical model (a theory) in order to understand what might be hidden underneath our ordinary understanding. This research method, though successful in hard science such as physics, proves disastrous in the study of human institutions for several reasons: (1) we cannot step outside of ourselves to study ourselves; (2) human institutions develop and change over time; and (3) (most important) all such attempts begin with and are colored by the researcher's political bent and assumptions or prejudices about the foundational nature of the institution studied. As will be discussed, hidden structure theories, such as those put forth by Kelsen, Rawls, Hart, Dworkin, and Unger cannot be verified and inexorably lead to nihilism.

---

[1] The Federalist No. 51 (Madison 1788c).

As shall be shown preliminarily through a necessarily abbreviated discussion of 1000 years of history, the Continental legal tradition regards law as something designed and created by governmental elites (i.e., by legislatures and bureaucracies), that is imposed on man by government in order to improve society, and it is properly grounded in deductive logic and (since the French Enlightenment Project) the scientific method/exploration.

In contrast, the common law traditionally regards law as developed over generations by a judicial custom of explication since supplemented by deductive logic and legislation produced by popular government, but legislation is assumed to be consistent with the common law and is itself interpreted by judicial explication. Common law traditionally perceives the purpose of law as a set of principles designed to protect peace; it holds that governmental power must be limited, and it posits that law should be non-instrumental (i.e., it is not aimed at improving society), but is aimed only at the practical management of conflict, that truth is most likely to be shown when adversaries battle each other before a disinterested judge, and that societal improvement is generated by cultural change and popular consensus that change is necessary, not from experts.

The two historical explications include detailed explanations of what their major legal philosophers have said about them, followed by a Discussion that compares them and a Conclusion. Some legal philosophers (Hart and Dworkin) who were educated in the common law tradition have adopted concepts that developed out of the positivist/analytical/continental movement of the 19th and 20th centuries. It is the author's thesis that the original common law legal approach as described by Lon Fuller is the one actually used in the Anglosphere and this is a more productive way to approach the study of how and why a particular legal system works—and where it fails; and the hidden structure methodology of Rawls, Hart, and Dworkin has led inexorably to less public trust in the law and a less civil society.

### 3. The Continental Tradition

*3.1. History from Ancient Greece through the Eighteenth Century*

The Greek philosophers Plato, Socrates, and Aristotle set the foundation for much of Western thought. To them, law was a general feature of the universe, the laws of physics and the laws of human society were both immutable, and ethics, politics, and religion were all one. Aristotle posited that every concept and every entity has a telos: a specific place in a deductive order or a specific role in society, that telos is immutable, and the state is at the top of that order. The state's telos, according to Aristotle, was to aim for the highest social good by imposing strict religious, social, and political discipline. Plato's view was somewhat different. He believed that an ideal society is never reachable in this world, but ethical ideals would help a polity choose between leaders. Thus, for him, the ethical defined the political.

Rome's conception of law was somewhat more developed and more nuanced. As Rome was falling, Christian Emperor Justinian in the Eastern Empire compiled his *Corpus Juris Civilis*, which stipulated that the emperor was the law. The last Roman emperors incorporated the Catholic church into Rome's bureaucratic machinery, and the Church developed legal institutions. As the Church's influence grew, it became the primary source of literate clergy, who rediscovered and translated Greek and Roman texts: Aquinas defined law as "an ordinance of reason for the common good, promulgated by the one who is in charge of the community." (Lesaffer 2009, p. 152). Aquinas adopted Aristotle's teleology and the belief that governmental elites should create law and that law should be grounded in syllogism and deductive reasoning. In contrast with Aquinas, Augustine of Hippo rationalized Christianity using Plato but separated between the City of God and the City of Man by arguing that the ethical must be detached from the political and the spiritual.

Charlemagne conquered western Europe, converted large populations to Christianity, and founded feudalism. With its growth, the tribal law-making assemblies of the Germanic tribes that had themselves conquered Rome disappeared, as did their conception that the law was something immutable and possessed by the people. Different localities had differ-

ent legal customs until Roman law provided some common ground among Continental societies. At the beginning of the twelfth century, the University of Bologna started teaching both canon law and Justinian's Digest, creating both church ('canon') and 'secular' lawyers. Roman-law-trained lawyers became much in demand (for complicated reasons), and more universities followed suit, training young men in a tradition that created a myth of kingship. The myth of kingship gave kings the powers to tax, mint money, conscript labor, and mete out justice. Kings needed centralized, bureaucratic governments to administer these powers, and thousands of young men saw the study of law as a pathway to economic stability. Roman law (*ius commune*) spread quickly (though unevenly) across the continent through these young, new bureaucrats. However, the Corpus Juris Civilis was already incomplete, archaic, and turgid when rediscovered in the twelfth century. To update and clarify it, glossators wrote explanations in the text's margins. That gloss quickly became bulky and unworkable.

In the thirteenth to fifteenth centuries, three events dramatically changed European law and society: Spain's conquering of the Western Hemisphere, the invention of the printing press, and the replacement of the glossators by humanists. As the Spanish conquest encouraged feudalism to be replaced by mercantilism, humanists such as Sir Thomas More and Erasmus spread their belief that a literate society would be better able to engage in civic life. They also developed two concepts: 1. the state as an entity independent from the church, and 2. the concept that there must be some sovereign entity that had the final word from which there would be no appeal. Sir Thomas More's *Utopia* (More 1516) was published in the Netherlands by his friend Erasmus and was widely read on the Continent (but not in England). In addition to criticizing feudalism, *Utopia* created a fictional state with no lawyers, no private ownership, and religious toleration. In *Utopia*, lawyers were unnecessary because laws were simple, and all social gatherings were public, pressuring participants to behave well. Furthermore, as all (men and women) were well-educated, there was no need for private ownership.

Context and background are necessary for assessing the role of government and law, and as shall be seen, the context in which the humanists were writing differed from that in England. More's *Utopia* did not accurately reflect the living conditions, customs, or law applicable to ordinary continental peasants, the largest population segment, who were organized into farm-based family households. Those farms' primary function was to provide for the family's subsistence needs.[2] A central feature of continental farms was that ownership was not individualized; the children were both farm workers and heirs to the farm, which was handed down from generation. So, an individual was only a temporary manager of the land that belonged to the family, even though it might have appeared on the surface that the eldest male was the owner (Macfarlane 1978, pp. 131–32). Women in marriage and younger male children were geographically immobile—they generally lived out their lives in the same village in which they were born or a neighboring one. Generally, the authority pattern within these households was patriarchal, and that authority was absolute.

Furthermore, women generally held a lower status than men and did not have any property rights if there were any living male family members. The gap between the peasantry and other social groups was very pronounced, and there was little, if any, mobility among them. In Eastern Europe, this pattern lasted into the nineteenth century.

While Thomas More was designing *Utopia*, Erasmus redefined the relationship between the individual and religion, arguing that because literacy was now widespread and the Catholic church's piety untrustworthy, individuals should themselves read the sacred texts and take individual responsibility for following Christ's example. Erasmus's work was a preview of the Protestant Reformation and the religious wars in much of Europe

---

2  (Macfarlane 1978, pp. 15–27). Macfarlane is describing eastern European peasantry (which lasted into the nineteenth century and consequently about which more is known than was recorded about western European peasantry) out of a well-reasoned belief that it was quite similar to that what which would have been found in western continental Europe.

during the sixteenth and seventeenth centuries, so much so that it was said that "Erasmus laid the eggs that Luther hatched." The result was that the Catholic church's power was weakened while European kings now claimed both divine ordination and absolute authority. While previously the power to make law was divided with the Church, now kings claimed that power, as when Louis XIV declared "*L'état, c'est Moi*." (In fact, kings never actually had absolute authority because their powers were limited by both their ministers and their pocketbooks, but they nevertheless claimed such authority. (Bouwsma 1988; Mettam 1991)) The Reformation and the rise of individualism necessitated a new view of the law.

The Natural Law movement initially divided law into three categories: 1. God's eternal law; 2. Natural law (principles implanted by God into men's minds and accessible to reason; and 3. Human law, which was derived from natural law (Pagden 2011). Descartes, licensed in canon and *ius commune* law before abandoning it, founded seventeenth-century rationalism in his work *Discours de la Methode* (see Descartes 1637) (A Discourse on Method). Descartes argued that the only proper and reliable method of attaining knowledge is deduction from first principles, and this approach became fundamental to all hard science as well as continental law (Hernàndez Marcos 2009, pp. 70–71). Natural law scholars Grotius and Pufendorf followed suit. Grotius, rejecting both *ius commune* and canon law, defined natural law as the rule and dictate of right reason, with the source of right reason ultimately being God, thus further separating the law of man from the law of God, while still endorsing kings' (and thus governments') absolute power (Grotius [1738] 1950, p. 250). Pufendorf pushed for the scientification of law and transformed natural law theory into an academic subject, arguing that human conduct can be governed by formal rationality and that private law should be organized into a rational, secularized system.

Though politics and law were one and the same before the seventeenth century, because of influential thinkers such as Descartes, Grotius, and Pufendorf, by the eighteenth century, the theoretical science of public law was based on deductive reasoning. It was now separate from the practice of politics (i.e., ways of making government effective) (Nedzel 2020, p. 62). In *Les Lois Civiles Dans Leur Ordre Natural* (1689), Domat [1689] (Domat [1689] 1850) formulated the pattern for most Civil Codes with sections on family law, property law, and obligations. His work tied Roman law, Christianity, and deductive logic together; for both Domat and Pothier, law was still a science built from centuries of experience. The French Enlightenment brought changes not so much to the substance of law (e.g., Napoleon's Code tied together the Roman heritage plus customary (local) law (Batiza 1984)), but it brought significant changes to the conception of law and the understanding of what constitutes a well-structured government—i.e., it led to the rise of democratic, constitutional republics.

The Enlightenment saw itself as presenting a new, clear vision of man and his relationship to the world, transferring belief in religion to faith in the new scientific reasoning. Enlightenment writings included constant metaphors to light or science. Instead of God, whatever question one had, nature and Cartesian reasoning would reveal the answer and lead to a more perfect society (Becker 1932, p. 53) To the French Enlightenment, preexisting legal systems were both irrational and dysfunctional because they lacked a unitary structure that could guarantee a uniform and equal administration of the law. The Philosophes saw chaos, dysfunction, and corruption in the multiplicity of administrative authorities (feudal, military, ecclesiastical, fiscal, local, royal) and in the judicial system. French courts' jurisdiction overlapped, causing contradictory decisions; they lacked a consistent procedure and legal authority. Judges were not held accountable for their decisions, and corruption in the judiciary was widespread. The Enlightenment view became that the proper judicial function was a mere mechanical application of appropriate law to given facts, not requiring any stated explanation of why or how the law applied or whether the purpose of the law would be accomplished by applying it.[3] This mechanistic view of how judicial decisions are

---

3　Nedzel (2020) at p. 64 and sources cited therein.

reached remains in civilian jurisdictions to varying extents—less so in Switzerland (because of its federal structure) and Germany—nevertheless, civilian judges are generally accorded a status similar to that of civil servants, not the exalted status they enjoy in common law jurisdictions. Civilian trial procedure remains judge-driven rather than adversarial.

Enlightenment thinkers on both sides of the English Channel endorsed what became the driving ideas of liberal culture: individual rights, the rule of law, republican (limited) government, toleration, and a free market economy (Capaldi 1998, pp. 349–51). However, they had two distinct and competing views of human nature. On the Continent, as initially posited by Helvétius and then endorsed by Bentham, the alleged essential truth about human psychology is that every individual is by nature governed by rational self-interest, and his or her response to environmental stimuli is to maximize pleasure and minimize pain. The Scottish Enlightenment, however, posited that enlightened self-interest implies that human beings can manage their own affairs without government interference and are responsible for their actions and the consequences thereof; thus, they posited a society of free competition and trust in the reason of the common man. Voltaire and the French Philosophes similarly adopted Locke's view that man should conquer nature but rejected the Scottish enlightened self-interest and instead supported a social technology that could solve all social and political problems. They eschewed limited government and, in its place, conceived of the following five political views:

1. Human beings are basically good, and the goal of human existence is happiness in this life (not in heaven);
2. The institutional practices most compatible with human happiness include the liberal culture of individual rights, market economies, the rule of law, and tolerance;
3. Human beings should be understood mechanistically: evil behavior is exclusively the result of external forces and the environment;
4. Social technology can control external forces and create a utopia;
5. Society is a hierarchical structure best served by a powerful and authoritarian state supervised by experts.

In place of absolute monarchy and its dysfunctional institutions, French Enlightenment thinkers proposed not just the liberty, equality, and fraternity of a democracy, but also the belief that human existence had a common destiny and therefore it was necessary that everyone be involved in politics. The *Philosophes* adopted (or thought they adopted) many of Rousseau's ideas: Law should be legislated in accord with Rousseau's *Volonté Generale*, the General Will, which he defined as the unanimous decision that people would reach if they properly ascertained the common good (Rousseau 1762, p. 24). Freedom, for Rousseau, was ridding oneself of considerations, interests, preferences, and prejudices, whether personal or collective, which obscure the objectively true and good. "The general will ultimately become a question of enlightenment and morality, a drive to create harmony and unanimity, so that the whole aim of political life was to educate and prepare men to will the general will without any sense of constraint. Human egotism must be rooted out, and human nature changed."

Rousseau believed that man must be forced (by the state) to regard himself not as a unique individual responsible only for himself, but as a being who functions in harmony with society (Talmon [1952] 2021, pp. 38–42, 48). Ideally, individuals join together through a tacit social contract, submitting themselves to the authority of the general will in a society of equals (Rousseau 1762). The state, when it has succeeded in disciplining mankind to comply with the general will, will have achieved its purpose. He argued that a government's duties included protecting its people, preventing the extreme inequality of fortunes by shielding citizens from becoming poor, keeping plenty within the reach of individuals, and remaining vigilant in restoring or maintaining patriotism and good morals (Rousseau 1755, p. 18). The existing laws, as Rousseau saw them, were an instrument the rich used to exploit the poor (Talmon [1952] 2021, p. 51). Rousseau's sovereign was the externalized general will; to become a reality, it must be willed by the people and if the people do not will it, then

they must be made to do so. (Though he never fully explained *Volonté Generale*) (Capaldi and Lloyd 2016, pp. 19–23).

Rousseau was very uncomfortable with personal property and the Scottish Enlightenment's belief in the free market (though that discomfort did not manifest in the early French republics), famously stating that "[t]he first man who, having enclosed a piece of ground, bethought himself of saying *This is mine*, and found people simple enough to believe him, was the real founder of civil society. From how many crimes, wars and murders, from how many horrors and misfortunes might not anyone have saved mankind, by pulling up the stakes, or filling up the ditch, and crying to his fellows, "Beware of listening to this impostor." (Rousseau 1755, p. 161).

Rousseau was also very concerned with inequality and believed that all forms of government (monarchy, aristocracy, democracy), were products of the differing levels of inequality in their societies and would lead to ever worse levels of inequality until overthrown by a revolution and the emergence of new leaders (Rousseau 1755, pp. 181–86). One can understand his concern, given the rigid class distinctions among France's three estates (aristocracy, clergy, and everyone else). A member of the third estate himself, one can understand the anger underlying the following quote attributed to both Rousseau's friend Diderot and Jean Meslier's *Testament* (1725): "Man will never be free until the last king is strangled with the entrails of the last priest."

The *Philosophes* generally held that legislation must then be enforced by the state and (simply) applied by the judiciary in accord with their understanding of Montesquieu's separation of powers. They also saw history as both universal and progressive. The 1789 *Declaration of the Rights of Man* shows the Enlightenment values of equality, liberty, property, security in one's person, and so on, rights that had not previously been recognized or enforced in France, but it was not clear that it was part of foundational law, because it was not legislated. It announces that "Men are born free and equal in rights. Social distinctions may be based only on common utility," starting with liberty, but focusing on equality and the expurgation of class distinction and less on liberty, consistent with Rousseau's *Social Contract*. In contrast, the 1776 *U.S. Declaration of Independence* starts with equality but quickly focuses on liberty: "All men are created equal," and the purpose of government is to protect the individual right to "life, liberty, and the pursuit of happiness."

In 1789, when Louis XVI summoned the Estates General in an effort to avert an economic crisis, many of the representatives of the third estate left the (unsuccessful) meeting to gather on the Tennis Court, and so the Revolution began. At first, a constitutional monarchy was created with law-making power transferred from the king to the legislative body with the King having a constitutional veto. However, the poorest were still disenfranchised in the bourgeois effort to create a balance between a new society and stability, and so finding that the Revolutionary purpose had not been fulfilled, the *sans-culottes* expelled the Girondists, leading to the Jacobin take-over and the reign of terror (Talmon [1952] 2021, pp. 78–79). The theory that drove Jacobinism was that the Revolution opened the way to a natural rational and final order of things; Robespierre was convinced that the people's will, if allowed complete expression, would prove identical with the true general will, a true, absolute, and universal morality. To Robespierre, the British system with its separation of powers was a fraud and a plot against the people. The Revolutionary aim was to extirpate tyranny altogether and let the people rule, "Let the people speak, for their voice is the voice of God, the voice of reason and of the general interest!" (Talmon [1952] 2021, p. 105). To the Jacobins, the general will was realizable only in the collective experience, and mere acquiescence was vicious egotism: "The factions are the most terrible poison of the body politic, they put the life of the citizens in peril . . . ; it is force that makes law . . . . In dividing the people the factious put party fury in place of liberty." (Talmon [1952] 2021, p. 116, quoting St. Juste)

The Jacobin constitution of 1793 guaranteed "to all Frenchmen equality, liberty, security, property, the public debt, free exercise of religion, general instruction, public assistance, absolute liberty of the press, the right of petition, the right to hold popular assemblies, and

the enjoyment of all the rights of man." (Capaldi and Lloyd 2016, pp. 61–62). However, it was never instantiated. Its democratic perfectionism with regard to plebiscitary approval of Legislated law and the people's right to resist oppression lead directly to anarchism and inverted totalitarianism based on a fanatical belief that there could be no more than one legitimate popular will. (Talmon [1952] 2021, p. 104). No dissent from the Republican government's view could be tolerated: not only traitors, but also the indifferent and the passive must be punished. (Talmon [1952] 2021, p. 114). Robespierre decided that the aim of his revolutionary government was to found a constitutional regime, but that could be established only in conditions of peace. As that did not exist at the time, France should be governed by Committee, and thus was created the anachronistically entitled the Committee of Public Safety (*Comité de Salut Public*) and the Terror began. (Talmon [1952] 2021, pp. 118–27). The instability ended only with Napoleon's rise, but the theories remained that the purpose of government is to improve people, and that law is created top-down by the government.

After the French Revolution, legislation became the primary source of law, as that was the closest method possible to replicate a general will. The Enlightenment (and of course Napoleon) led to the proliferation of Civil Codes, designed to be a body of private law logically and rationally organized and universal in nature, so it was portable. France's Civil Code was based on a combination of local customary law and Roman law, but different countries developed similar Codes. The codification movement was heavily influenced by the ideas of the times, initially those of Jeremy Bentham. They were adopted by legislatures and replaced all previous laws. As they and other legislated laws were the solemn expression of the legislated will, they could not be changed or avoided other than by the legislature itself. No independent review was necessary or allowed, as they represented the General Will.

While many things changed because of the Enlightenment and some things have changed since the Enlightenment, concepts that have persisted in the Civilian Tradition include the focus on rationality and a scientific approach to law, the concept that legislated law is the expression of the people's (general) will and a collective good, and the belief that the purpose of government and law is to control and improve society.

### 3.2. The Rise and Fall of Rechtsstaat

*Rechtsstaat* or *L'État de Droit*, which is most accurately be translated as *Rule Through Law*, describes the relationship between the state and humankind in the Civilian Tradition. Immanuel Kant, though he did not use the term, is regarded as the spiritual father of the concept as he defined the state as the union of a multitude of men under laws that are grounded in reason and which protect equality, freedom for all, and individual autonomy (Heuschling 2002; see Kant's Social and Political Philosophy 2022). Kant's conception was that man should never be treated merely as a means to an end, and the government's powers should be limited so as not to interfere with individual freedom—i.e., what is now described as negative rights. However, the concept as popularized by Robert von Mohl in 1844 differed: Von Mohl believed that the state should comply with the law and the state's purpose was to promote an individual's complete development. Under this conception, governmental authority had no specific limitations, and its primary purpose was to serve the people's interest by promoting positive, listed rights. Von Mohl's vision was in turn superseded by Georg Jellinek and R. von Jhering, who founded *rechtsstaat* on three concepts: the state must limit its own powers, rights were subjective in that the state established them by means of its authority (they are not inalienable to the individual), and the state has primary status. Individual rights were neither of a pre-political origin, nor of a religious nature, nor based on any universal natural law.

Jhering, like many legal scholars of the late eighteenth and early nineteenth century, was influenced by the utilitarian movement, founded by eighteenth-century philosopher Jeremy Bentham (DiFilipo 1972). Despite being English, Bentham endorsed the Continental, specifically French, intellectual tradition. A long-standing critic of Blackstone and English

judge-made law, Bentham strongly supported the codification movement because he believed that a Code would make law complete, internally consistent, simple enough for a common man to understand, and universal, a goal consistent with that of the Philosophes. However, he is primarily known as the founder of utilitarianism and positivism. The utilitarian principle is that an action is right insofar as it promotes happiness, and the greatest happiness of the greatest number should be the guiding principle of conduct (The History of Utilitarianism 2014). Bentham sharply criticized natural law as fiction because he argued that it blurred the distinction between law as it is and law as it ought to be (Hart 1958).

Jhering, known as the "German Bentham" (Von Jhering [1913] 2009, pp. xix–xxii, 33, 409) assumed that law is and should be instrumental, even titling his work to that effect—in translation, "Law as a Means to an End." Jhering's work, like Bentham's, was grounded in utility, but he rejected both Locke's (and Blackstone's) consent theory of law and what he saw as Bentham's individualism, arguing instead that his own views were based on psychology, that the state does and should use a system of punishment, reward, and coercion to improve society, and that man owes duties to the state. The Continental attitude toward Anglospheric law, at the turn of the 20th Century was that the Lockean consent theory of government was "wabble."

In addition to the consent theory being "wabble," the Vienna Circle in their 1929 *Postivist Manifesto* explicitly asserted that a scientific world view should influence the forms of personal and public life, in education, upbringing, architecture, and the shaping of both economic and social life (Sarkar 1996; Discussed in Nedzel and Capaldi 2019, p. 169). For positivists, normative statements (statements of morality) are not empirically true. At best, they are expressions of subjective preferences. Thus, all traditional sources of moral authority are illegitimate, including Christianity and its theology of natural law because statements about God or religion are neither empirically true nor empirically false and therefore can be dismissed. Even secular versions of natural law theory presume some sort of universal human telos can be dismissed because they cannot be empirically proven.

Positivism spread among German theorists, but there was no agreed-upon understanding of it, until Hans Kelsen, who had studied Jellinek's work, denounced both post-natural law theories and *rechtsstaat* as shams and political values that would eradicate the science of law—and indeed, by the time of the Weimar Republic, *rechtsstaat* was so formalized and denatured by positivist theory that one could accurately have described it as a magic box out of which a jurist could produce anything wanted (Heuschling 2002, p. 154). In 1934, Kelsen first published his *Pure Theory of Law*, wherein he tried to explain the universal nature of law objectively. In it, Kelsen described law universally as a science (though distinct from physical science) based on a pyramid of norms (not moral principles), culminating at the top in a hypothetical norm that he termed a *Grundnorm*. As Kelsen described it, a *Grundnorm* is a foundational postulate that is assumed, an epistemological choice that provides an understanding of the legality of a state's constitution and therefore of its entire legal order. Thus, Kelsen's work was an attempt to deduce the law's universal hidden structure, paralleling the kind of explication that is used in nuclear physics and other hard sciences. Every legal system has its one basic norm, according to Kelsen (Raz 1974). Kelsen's structure, in which an inferior norm cannot contradict a superior norm, means that if an inferior norm (or law) contradicts a superior norm, then the inferior norm must be corrected or removed.

During the time Kelsen started writing and before he was removed from his professorship at the University of Cologne by the Nazis in 1933, the Weimar Republic (1918–1933) governed Germany problematically: stiff reparation payments following the First World War led to hyperinflation along with high unemployment and social and political unrest.[4] Moreover, the constitution had serious weaknesses. It did not ban political parties whose aim was to overturn the constitution; it listed a number of rights, but they were expressed

---

[4] "The Weimar Republic" and "Article 48" in the Holocaust Encyclopedia (U.S. Holocaust Memorial Museum) at https://encyclopedia.ushmm.org/content/en/article/the-weimar-republic (accessed on 25 September 2022)

in terms of principles, not inviolable rights that the government was obligated to enforce, and it set the bar for a no-confidence vote very low, which led to frequent dissolutions. The provision that proved disastrous, however, was Article 48 which gave the president the power to take any step necessary to restore order and defend the German people. The public's political and economic dissatisfaction led to the rise of the National Socialist German Workers' Party (the Nazi party), which then used Article 48 to establish Adolf Hitler's dictatorship.

While the horrors of World War 2 led to the fall of the Nazi party and a world-wide concern that future republics be insulated against genocidal dictators, legal positivism remained prominent. As H.L.A. Hart, influenced by Kelsen, described legal positivism: (1) laws are commands made by and enforced by human beings (a rejection of natural law); (2) law and morality—the *is* and the *ought*—are not necessarily related; (3) the study of law and its (hidden) structure (i.e., the theory of law) differs from the study of the history or sociology of law or the study of the function, aims, or moral value of law; (4) a functional legal system is one where correct decisions are deduced from predetermined legal rules; and (5) unlike law, moral judgements cannot be established or defended by rational argument (Hart 1958, pp. 601–2). The political dimension, not the moral dimension, is the basis of law.

*3.3. Post-World War 2, Rechsstaat Is Reborn, as Is Legal Philosophy—Or Is It?*

After World War 2, Germany reinvented itself, including specific values in its foundational documents, foundational principles that had not previously been stated. Kelsen's work convinced a number of law makers that a judicial review of legislation was necessary to make sure that new legislation was within constitutional boundaries, and that in fact such review was authorized by the innate deductive structure of law. In fact, Kelsen included a form of judicial review in the constitutions he drafted for Austria and Czechoslovakia before Hitler took power (Wolfe 1994). After World War 2, a number of countries similarly created constitutional courts and looked for other ways to instill limitations on governmental power. Germany's federal constitutional court held that the new German Fundamental Law (*Grundgestez für die Bundesrepublik Deutschland*) included *rechtsstaat*, which is now defined as including two components: 1). A formal *rechtsstaat* which focuses on guarantees of legal supremacy and checks on state power, and the substantive *rechtsstaat*, which guarantees fundamental values such as basic rights (Heuschling 2002, p. 154). The concept of self-limitation as well as the concept that it is the government, a political entity, that grants individual rights, as developed under Jhering, has been eliminated, with the intent to return to the concept as originally developed by Kant.

However, tension between the Fundamental Law's socialist values as stated in Arts 20 and 28 of the Basic Law and individual liberties remains. Article I mandates that while the state must protect human dignity and human rights, a citizen owes duties to government and society, duties that implicitly limit individual liberty. The Fundamental Law requires that the state provide social welfare benefits to remedy social inequality and "balance or correct the unfortunate effects of a market economy." It also provides that individuals are both dependent on and committed to the community; thus, while the state must guarantee and nurture an individual's dignity, that is within the constraints of social solidarity and responsibility. Thus, these concepts show an inheritance from Rousseau's (and Marx's) discomfort with market economies, as well as a continued acceptance of a version of his General Will and its manifestation as being expressed in legislation. Law is still assumed to be something developed by the government and governmental experts and which should be part of a coherent, deductively organized system.

*3.4. Post-World War 2 Positivism—Kelsen to Hart*

By the late 19th century, positivism and the progressive movement were influencing American jurists. Realists such as Oliver Wendell Holmes, Jr. seemingly rejected positivism, arguing that "the life of the law has not been logic: it has been experience" (Holmes 1881),

but they nevertheless unquestioningly accepted legislative supremacy, a view that reflects positivism more than it does the common law tradition that typically looked on legislation with suspicion and interpreted it narrowly.[5] Post-World War 2, as will be discussed in Part 3.2, Lon Fuller rejected legal positivism as providing a mechanistic and formalistic vision of legal reasoning, and that the positivist insistence on the distinction between *is* and *ought* shows an indifference to the moral status of law, both of which led to Germany's missteps. However, Oxford Don H. L. A. Hart (1907–1992) rigorously rejected this argument in a long series of famous debates with Fuller, a Harvard law professor, whom he (mistakenly) assumed was promoting traditional natural law.

Hart's positivist theory of law had little in common with the Lockean view. Instead, using exploration to hypothesize the hidden structure of law, Hart's position reflects the Enlightenment Project's view of the relationships among law, government, and society:

1. Human beings are basically good, and their ultimate goal is happiness in this life;
2. The goal of legal philosophy is to derive institutional practices that are most compatible with liberal culture;
3. Human beings are to be understood mechanistically, i.e., evil is exclusively the result of a corrupting environment;
4. Social technology can create a utopia by controlling that environment;
5. Society is best served by a powerful, authoritarian state run by experts (Capaldi 1998, p. 351).

Hart's best-known work, the *Concept of Law* (Hart [1961] 2012), combined Kelsen's theories and views with Jeremy Bentham's, thus continuing the expansion of legal positivism. Though Hart's and Kelsen's theories differ somewhat about the nature of law, they both developed theories about a universal hidden structure of law. While Kelsen insisted that positive law theory should be limited to jurisprudence itself, Hart expanded it to incorporate ideas from philosophy and sociology (Culver 2001). H.L.A. Hart studied classics at Oxford, became a barrister, and practiced for 8 years before World War 2 during which he served with British military intelligence and became interested in philosophy through ties at Oxford. After the war he accepted a teaching fellowship in philosophy at Oxford, and from there was elected Professor of Jurisprudence and given a fellowship at University College, where he wrote *The Concept of Law*. Unlike Kelsen, he did not posit a structured pyramid of law, wherein every law has a place. His theory about the hidden structure of law and legal positivism was simpler but still structured: he posited that there is a distinction between primary and secondary legal rules present in every legal system. Primary rules govern conduct, while secondary rules govern procedural methods by which primary rules are enforced (Hart [1961] 2012). Hart posited that there are only three secondary rules: (1) a "Rule of Recognition"—the rule by which any member of a society may discover what the primary rules are and be assured that they are legitimate (echoes of Kelsen's *Grundnorm*); (2) The Rule of Change—the rule by which existing primary rules might be created, amended, or deleted; and (3) The Rule of Adjudication—the rule by which a society determines when a rule has been violated and allocates a remedy. He divided primary rules into two kinds: rules that delineate duties, and rules that grant powers.

Hart still adhered to the positivist view that that law may, but does not necessarily, adhere to conventional morality (Hart [1961] 2012, pp. 185–86). Writing during the upheaval of the 1960s and its conflicts between individual freedom and legislation aimed at wealth redistribution or freedom of expression and concerns about maintaining public standards of decency, Hart argued a distinction between the existence of legal or constitutional liberty and its value to individuals (Hart 1955, p. 53). So, under this view, a law securing freedom of the press that is written so as to be universal does not have a universal effect because it is meaningful only to those who are literate or who have access to books and newspapers, or who have the time to read them (MacCormick 2008, pp. 21–22). Those who own or control newspapers and publishing houses or the media have even more access to this freedom.

---

5 See discussion of legislative supremacy infra lines 1271–1276.

Thus, Hart was arguing that classical liberalism should be revised by superimposing on it a social democratic strategy aimed at narrowing the inequalities in the actualization of liberty—very much a socio-democratic viewpoint, one that implies that the purpose of law and government is to improve society and is therefore consistent with the civilian tradition. Consistent with the positivist tradition, he separates law from morality, saying that any morality in law is brought in through politics because morality is subjective and cannot be scientifically derived, therefore it should be left to politics and excluded from any exploration of a legal system.

*3.5. Analytic Philosophy, Rawls, and Dworkin*

As the 1960s faded into the 1970s, positivism was replaced by analytic philosophy, a related view, as exemplified in Rawls' and Dworkin's writings about the philosophy of law. This view recognizes norms as substantive entities instead of political chimera, but those norms cannot be factually established, and the hidden structure of norms cannot be separated from substantive political views (Capaldi 1998, pp. 367–72). For example, John Rawls (1921–2002) in "A Theory of Justice" (1971) explored the alleged hidden structure behind justice in an effort to modify people's preconception of what it means, calling his method "reflective equilibrium." (Rawls 1971, pp. 46–53). Rather than explicating common experiences that individuals describe as "just" or "unjust," he leaves meaning and definition questions aside to develop a "substantive" theory, starting from a hypothetical neutral position that he describes as "behind a veil of ignorance."

Rawls was aware of the divide between Locke and Rousseau and brought normative thinking back from its positivist exile (Capaldi and Lloyd 2016, pp. 172–75). According to Rawls, no one deserves what they have regardless of how they acquired it (thus dismissing Locke) because we are all entirely products of the genetic lottery and historically accidental family circumstances into which we were born, thus the resentment children feel (and which Rousseau discusses in *Emile*) is justified (Rawls 1971, p. 540). As that is the case, we must have some form of redistribution because otherwise the powerful will always impose on the weak. So, starting with Rousseau's assumption that no social order is legitimate unless founded on an original unanimous consent to procedure, Rawls postulated his hypothetical original position in which individuals, entering society naked and giving up all claims to previous property and advantages, choose principles of justice from "behind a veil of ignorance." He posits that everyone's well-being is dependent on cooperation without which no one could have a satisfactory life.

Rawls claimed his work was informed by Kant's, but his view of individualism differs significantly from Kant's. Rawls, like Kant, argues for the primacy of liberty in his two principles of justice. However, Kant argued not only for fundamental human autonomy, but he also argued that redistribution treats persons as a means for the good of others (rather than as an end in themselves), and he argued against determinism and the view that justice was in any way concerned with self-fulfillment (Capaldi 1998, pp. 369–72). Rawls contends that the most important primary good is self -esteem and that self-esteem depends on our seeing ourselves through the eyes of others, which is the exact opposite of Kant's view of individualism and personal autonomy.

Rawls further posits that there is a pre-determined understanding of what is good for society, and that is justice, which is purely procedural to "nullify effects of special contingencies" that often encourage people to exploit others for their own advantage, thus causing discord. The further implication of Rawls' view is that all individuals, under the appropriate conditions, will make choices that lead to a cooperative and peaceful society—in essence, he restated Rousseau's General Will. He developed two principles of justice: (1) each person is to have an equal right to the most extensive total system of equal basic liberties compatible with a similar system of liberty for all, and (2) social and economic inequalities are to be to the greatest benefit of the "least advantaged." The latter he describes as the Difference Principle, which is a substantive end to be achieved by eliminating diversity (Capaldi 1998, pp. 369–72), and which is a response to socialist

concerns about equality. In other words, for both Hart and Rawls, the government's role is to "equalize" inequalities in liberty and redistribute wealth to help the downtrodden.

Modern French economist Thomas Piketty in his 2014 highly influential book, *Capital in the Twenty-First Century* supported Rawls' "difference principle." He starts his book with a quote from the French 1789 *Declaration*: "Social distinctions can be based only on common utility," that the purpose of the *Declaration* was to achieve a just social order, and he argues that high degrees of inequality are unjustifiable, quoting Rawls' in a footnote (Piketty 2014, p. 631, fn. 21): "Social and economic inequalities . . . are just only if they result in compensating benefits for everyone, and in particular the least advantaged members of society." Thus, though Rawls intended to prioritize liberty, ultimately, he (like Piketty) believes that justice lies in remedying societal inequalities.

Dworkin justifiably argues that Rawls never established the priority of liberty, and that "damage to self-respect that comes from seeing others better off in the social structure is such a malign influence on personality that people at the bottom can't really be better off overall, even if they're materially better off."[6] As Robert Nozick says, implementing Rawls' theory would require continuous governmental interference in the lives of individuals, and is thus counter to the very liberty Rawls claims to support.

Ronald Dworkin (1931–2015) is the next and best known of the Analytic (legal) Philosophers. He graduated from Harvard summa cum laude in philosophy in 1953 and was granted a Rhodes scholarship. Hart read his final exams and found them extremely impressive. Dworkin then returned to Harvard for law school, graduating in 1957, and clerked for the famous progressivist judge Learned Hand on the United States Second Circuit Court of Appeal. He worked for a major law firm in New York City for a few years, and joined Yale's law faculty in 1962, leaving it in 1969 when Hart named him as his successor at Oxford.

Dworkin began his tenure at Oxford by criticizing Hart's work. In *Taking Rights Seriously*, Dworkin rejected legal positivism (Dworkin 1977, pp. 9, 22; see also Dworkin 1986, pp. 34–35, 109, 431 n. 2). He argued that Hart's positivist theory does not consider the quandaries that judges face (Dworkin 1977, p. 22). This first work of Dworkin's claims to set forth a general theory of law. It does not do that but instead discusses and rejects various formulations of positivism, distinguishing among concepts such as "normative" rules and social rules, and it discusses the various ways courts should approach hard cases, using a fictional judge, Judge Hercules, to demonstrate. (His doing so is a clear parody of Lon Fuller's fable of King Rex, whom Fuller used to illustrate eight minimal conditions rules must meet in order to count as law (Fuller 1969). Dworkin roots law in morality, but it is abstract morality as determined by judges (Hercules) informed by integrity instead of custom, precedent, and other such standards. For Dworkin, a true theory of law is a theory of how cases ought to be decided, beginning with an abstract ideal about the conditions under which governments may use coercive force.[7] The norms are still detached from past practice; thus for Dworkin, the law ultimately remains "what the judge says it is."

In *Law's Empire*, his next work still focused on Anglo-American judge-made law, Dworkin again rejects the common law argument that judicial decisions must be grounded in prior reasoning, arguing that judicial "reinterpretation" of prior reasoning cannot be (scientifically) verified as being drawn from prior practice[8]—despite the common law habit of including a multiplicity of citations, all of which a reviewing court checks to verify that they accurately represent established law!

In place of Hart's theory, Dworkin claimed that the controversy is really about morality (Ibid., p. ix). Consistent with positivism, he posits that law and morality are not synony-

---

6    Dworkin quoted in (Magee 1982, p. 223).

7    "Legal Positivism" (Stanford Encyclopedia of Philosophy Plato.stanford.edu, https://plato.stanford.edu/entries/legal-positivism, accessed on 24 September 2022).

8    (Dworkin 1986, p. 6). Actually, at American common law, under professional rules and penalty of censure, all judges must cite the exact precedent on which they are basing their decision, and those citations are checked for accuracy by reviewing judges and their law clerks, who are themselves also attorneys. So, contrary to Dworkin's assertion, judges' reasoning not only *can*, but also *is* habitually confirmed as being drawn from precedent.

mous, in part because what people posit as moral is at times immoral, so in truth he is modifying positivism, not rejecting it. Dworkin focuses on two principles of political integrity in describing the proper way to make and view authoritative law: a legislative principle which requires that lawmakers make the total set of laws morally coherent, and an adjudicative principle, which instructs that the law be interpreted as coherent to the greatest extent possible (Ibid., pp. 176–77). He argues that this "integrity" would not be needed in a utopian state, but because government officials do not always do what is just and fair, integrity can require us to support legislation that might be inappropriate in a perfectly just and fair society and to recognize rights we do not believe people would need to have guaranteed in a utopia. Thus, like Hart, he justifies legislation that "puts a foot on the scale" in order to realize what he postulates as fair and equal liberty.

Using his "law as integrity" theory, Dworkin posits that judges should assume that the law is structured by a coherent set of principles about justice and fairness, and that they should enforce these two principles in every case that comes before them (Ibid., p. 243). Thus, both Dworkin and Hart, in keeping with civilian tradition, regard government (and law) as institutions that should be used for what some governmental entity perceives as the betterment of society, i.e., an enterprise association (in Oakeshott's terms).

The next question, therefore, concerns the nature of justice and fairness according to Dworkin. Dworkin emphasizes that he sees the underlying principle of Western law to be a "right to equal concern and respect," which supersedes the "so-called right to liberty." (Dworkin 1977, pp. 180–83, 272–78). Following up on the meaning of phrase, does equal concern and respect refer to equality before the law or economic equality, according to Dworkin? (Nedzel 2020, pp. 138–39). Does it refer to law as instrumental or non-instrumental? On these issues, Dworkin is initially ambiguous. At first, in *Taking Rights Seriously*, he seems to refer primarily to equality before the law: "the doctrine of precedent serves equality of treatment before the law."(Dworkin 1977, p. 37) However, when talking about rights, he accepts collective goals such as economic redistribution, consistent with his criticism of Rawls: "a community may aim at a distribution such that maximum wealth is no more than double minimum wealth;" and "[g]overnment must, of course, be rational and fair; it must make decisions that overall serve a justifiable mix of collective goals . . . ." This acceptance of instrumental, collective law—particularly when he discusses his rejection of the implied-consent view of government—shows Dworkin's view of law as something aimed at improving society and thus in line with traditional civilian thought (and oriented towards socialism): "our aim . . . is to develop a theory that unites our convictions and can serve as a program for public action." (Ibid., p. 175).

Dworkin's views are also consistent with Germany's view that citizens have political obligations: "Integrity . . . insists that each citizen must accept demands on him, and may make demands on others, that share and extend the moral dimension of any explicit political decisions. Integrity, therefore, fuses citizens' moral and political lives . . . ." (Dworkin 1986, p. 189). In order to take rights seriously, we must first have a deep theory of "human dignity" and "political equality." Over time, Dworkin clarified his belief that a legal system must reflect a community's "particular overriding goal," and that this collective goal (echoes of Kelsen's *Grundnorm*) trumps individual liberty. Moreover, he makes clear his preference for a government focused on equality qua economic equality/redistribution in his last work, *Sovereign Virtue*:

> Equal concern is the sovereign virtue of political community—without it, government is only tyranny—and when a nation's wealth is very unequally distributed, as is the wealth of even very prosperous nations now is, then its equal concern is suspect. For the distribution of wealth is the product of a legal order: a citizen's wealth massively depends on which laws his community has enacted—not only its laws governing ownership, theft, contract, and tort, but its welfare law, tax law, labor law, civil rights law, environmental regulation, law, and laws of practically everything else. (Dworkin 2000, p. 1)

Although Anglo-American educated, Dworkin's beliefs are consistent with the "hidden structure" approach that developed in the Civilian legal system and more specifically with analytic philosophy. While he does not discuss a deductive system of legal principles, he believes that the differentiation between law and everyday morality is important. Moreover, he believes that an overarching theory of law is important and that the proper purpose of law (and government) is to improve lives, with an emphasis on economic (as well as political) equality—a concept still consistent with the Ancient Greek's understanding of law as an instrument to improve mankind. Finally, without any explicit reference to Kelsen, he *de facto* posits that equality should be the *grundnorm* of every legal system with integrity and imbues his views with his underlying socialistic political bent.

In sum, the Civilian Tradition believes strongly that law should be drafted by experts and adopted, interpreted, and enforced by experts, that the purpose of law is to improve society, and that government creates law. Since the Second World War, the civilian tradition has focused on government's purpose being to enforce rights and remedy inequities of various kinds and has recognized that the government's power should be limited. In the period following the War, positivism first reached its heyday under such thinkers as Kelsen and Hart, to be replaced by a related concept driven by analytic philosophy, as exemplified in Rawls' and Dworkin's work. Positivism and Analytic Philosophy have been developed and adopted by legal philosophers on both sides of the Atlantic. Regardless, these two approaches both rely on the creation of scientistic theories and exploration, not on explication. Both approaches invariably end up finding that the ultimate value or *grundnorm* of Western legal society is equality, and the kind of equality that such thinking easily and usually resorts to as the proper aim of a legal system is to remedy economic inequality, as it is quantifiable and verifiable. However, as neither approach can be verified as true, the logical extension of both is the nihilism that characterizes Critical Legal Studies and Critical Race Theory, the next approach developed in and taught by the legal academy.

### 3.6. *Critical Legal Studies/Critical Race Theory/Critical Feminist Theory/etc.*

Critical Legal Studies is a logical reaction to scientistic thought such as Dworkin's and was the next movement to gain popularity. It is probably at its peak now in American law schools and elsewhere. In 1977 a group of academics gathered at Harvard Law School to denounce the theoretical underpinnings of American jurisprudence, objecting to Legal Realism, Formalism, Liberalism, "and everything else." Their commitment is to shaping a society based on some "substantive vision of the human personality, absent the hidden interests and class domination of legal institutions." Thus, the CLS movement considers all legal institutions and legal concepts to be illegitimate, having been promulgated by those in power against minorities of all kinds. Its most well-known advocate, Roberto Mangabiera Unger, acknowledges the wide diversity of thinking encompassed by the CLS movement, but states that what is shared in common is that all CLS views challenge society to consider the validity of its own institutions and reconsider the past "ultimate answers" upon which those institutions are based (Turley 1987).

As one might expect, the most direct philosophical antecedent of the CLS movement is critical Marxism founded by the Frankfurt School as an alternative to scientific Marxism: critical Marxism abandoned the scientific Marxist belief that socialism would develop gradually as capitalism inevitably fell. Critical Marxists such as Lukacs, Gramsci, and Sartre argued that a revolutionary consciousness can be achieved without waiting for Marx's theoretical incremental collapse.

A dominant theme running throughout CLS scholarship is the belief that legal autonomy is impossible, and ALL legal concepts are the result of exploitative dominance by those in power, and thus that it is all illegitimate. With regard to the positivist/realist/analytic philosophy view that law should become more "scientific," CLS proponents claim that the attempt to move away from what Hart/Dworkin viewed as the inherent bias of judges to a data-oriented, scientific approach using "objective" experts was simply a substitution of "technocratic consciousness . . . to defend the status quo without basing its policy choices

on some utopian vision of the good and just life." They similarly reject with disdain the formalistic, common law view that law as developed over time is fair and impartial and void of politicization, and they reject the classical liberal belief that neutral adjudication is essential in order to protect individual liberty against majoritarian power, charging that this is based on a false vision of "human sociability": the very act of legal interpretation is necessarily contextual and thus value-laden, thus it operates to resist any social change that would alter society's hierarchical structure.

Critical Race Theory is a version of CLS focused on using race and experiences of race to explain social, political, and legal structures and power distribution. Proponents of CRT argue that racism and disparate racial outcomes (such as the high rate of incarceration among African-Americans in the United States) are caused by complex, changing, and often subtle social and institutional dynamics rather than explicit and intentional prejudice on the part of individuals.

They hold that classical liberalism is incapable of addressing fundamental problems of injustice in American society despite the civil rights legislation and judicial decisions in the 1950s and 60s because its emphasis on the equitable treatment of all races under the law renders it capable of recognizing only the most overt and obvious racist practices, not those that are relatively indirect, subtle, or systemic (Encyclopedia Britannica n.d.). CRT proponents are dedicated to applying their understanding of the institutional nature of racism to the goal of eliminating all race-based and other unjust hierarchies, but do not agree on how to do this other than publicizing their claims of racism, nor do they have a vision of what the ultimate result should be.

They begin with the self-evident statement that race is a social construct with no biological basis, but then posit that racism is the ordinary experience of most people of color, an unproven claim. They argue as proof of the institutionality of racism that people of color are on average more likely to be denied loans or jobs than similarly qualified white people, that they are more likely to be unjustly suspected of criminal behavior by police and more likely to be victims of police brutality, are generally imprisoned more often and for longer periods, that they continue to live in impoverished neighborhoods, that predominantly Black and Hispanic neighborhoods receive fewer and inferior public services, including a lack of quality education and inferior medical care.

Supporters of the CRT movement further claim that legal advances apparently intended for people of color tend to serve the interests of dominant white groups instead. As initially argued by Harvard's Derrick Bell, the U.S. Supreme Court's landmark Civil Rights decision, Brown v. Board of Education (1954) was not actually aimed at eliminating segregation in the United States but was in fact was the product of a secret agreement between the U.S. Department of Justice and the U.S. State Department to improve the country's global image, a product of what CRT proponents call "interest convergence." In addition to "interest convergence" evidencing institutional racism, CRT supporters claim that minority groups undergo "differential racialization",i.e., they are periodically stereotyped depending on the interests or convenience of white Americans. Thus, before the 1960s civil rights protests, they were stereotyped as simpleminded and content with segregation; afterwards, they were viewed as natural-born criminals or leeches living off social welfare programs. Like the Jacobin's belief that dissenters should be sent to the Guillotine, CRT supporters believe that nonbelievers are racists and enemies and so no conversation with them is necessary and they can be "cancelled," i.e., excluded from any interaction (Krasne 2020). CRT concepts are interwoven throughout the thinking and actions of the Black Lives Matter and ANTIFA Movements, which led to a great deal of violence in 2014–2019.[9]

CRT is highly influential in American law schools and universities but at the same time is facing strong societal criticism. Critics argue that it is not based on fact, that

---

[9] (Onwuachi-Willig 2022). Ultimately, it has been shown that the founders of the BLM movement corruptly used the money they were given to purchase mansions for themselves, not to advance their movement.

it is dividing the United States into groups of oppressors and victims, thus increasing rather than decreasing racism; that it is infecting everything from politics and education to the workplace and the military; and that its skeptical and incoherent attitude towards objectivity and truth is nihilistic, dysfunctional, and destabilizing.

One can also argue that CRT assumes the existence of that which it attempts to prove, such things as systemic racism, white privilege, and selective police brutality even in the face of statistics that disprove those claims. For example, award-winning Harvard Professor of Economics Roland Fryer, J.R., who is himself Black and grew up in a poor urban environment, faced a firestorm of criticism for publishing a working paper in which he related statistics that surprised even himself concerning the use of force interactions between people of color and police. His controversial study showed that while Black and Hispanic people were 50% more likely to have interactions with police involving non-lethal, non-shooting use of force than were White people; however, Blacks were 27.4% LESS likely to be shot at by police, thus he concluded that he found no evidence of racial discrimination in officer-involved shootings, implicitly questioning the CRT/BLM demand to defund police. Later that same year he was apparently "cancelled": Harvard determined that he had fostered a sexually hostile work environment in his lab based on some off-hand remarks he made, suspended him without pay for 2 years, closed his lab, and barred him from teaching or supervising students (Casselman and Tankersley 2019). Since then, he has not followed up on the Use-of-Force research, nor has anyone else. Given the firestorm he faced, it is not surprising that no one else is willing to risk their academic career on such potentially politically unpopular research.

Heather MacDonald, a respected conservative commentator, attorney, and author, in her study of the BLM movement, similarly challenges the resulting crusade against law enforcement. Her research found that lies about what happened in some of these incidents are widespread and uncorrected by the media (e.g., Michael Brown was not gunned down in cold blood in Ferguson, Missouri); and that as a result of the BLM movement, officers no longer patrol assiduously, and criminals have become emboldened. Furthermore, she asserts that criminogenic environments in Chicago and Philadelphia show that black crime is not, in fact, the result of poverty and inequality; and the mass-incarceration disproportionality is actually the result of widespread Black-on-Black violence, not racism (MacDonald 2016).

Whether based on truth or not, possibly the best way to test the value of the CRT approach is to see whether its methods and mandated legal changes work. Most recently, in 2020 New York State instituted a "no cash bail" and immediate release policy for a number of listed crimes (including manslaughter, criminally negligent homicide, making terroristic threats, arson, criminal possession of a gun, grand larceny, criminal possession/sale of a controlled substance, resisting arrest, hindering prosecution, etc.), in the belief that minorities were being disproportionately charged (Quinn 2022). Since that time, because of a rash of high-profile crimes in subway stations and tourist hubs, three quarters of New Yorkers identify crime as a very serious problem, although most crime still occurs in poorer neighborhoods, overall numbers may actually be lower than twenty years ago and gun crimes have decreased. Nevertheless, overall crime went up 59% in the spring of 2022, with crimes on the transit (i.e., subway and train) system soaring by 73% (Akinnib and Wahid 2022; Eyewitness News 2022). Crimes for which judges were no longer allowed to set bail increased by double digits in the first $2\frac{1}{2}$ months after the reform took effect, 69.7% of the defendants arraigned on felony charges who were so released had a prior conviction or a pending case, and 30% of those same defendants were rearrested while their case was pending. The overwhelming majority of the victims of crime in NYC are Black and Hispanic, and the crime wave affects their neighborhoods disproportionately. Thus, these CRT/BLM-driven changes (and others like them in other cities) are of questionable efficacy.

Other CLS-driven changes that have proven controversial include the teaching of CRT and a cultural redefinition of gender from male and female to a host of created alternatives (despite the biological reality of only two distinct sexes) from kindergarten through high

school using new school policies, practices, and curricula (Ingraham 2021). Some states have passed laws against these curricula (e.g., Florida, Tennessee), others have adopted them (e.g., California), some parents have sued school districts (Virginia, Tennessee), and some parents have vociferously objected in school board meetings. For example, white parents in Virginia objected to the following excerpt from teaching materials, arguing that it incorporates CRT and discriminates against white people (Dorman 2021): "Since white people are in a state of privilege with regards to Racial issues (meaning they can choose not to think about racial issues that don't affect them), they may respond to the whole discussion of Race with discomfort." Black parents have objected to CRT materials as well. The following is a partial transcript of what parent Kayla Dunn of Idaho stated in addressing her school board:

> I represent myself, my family, I have 5 children ages 2–15 years of age. And I have sat to listen to all the CRT hearings, and I just thought it was time for me to speak up as a woman of color. And so, I thank you so much for the opportunity to speak to all of you today and to make my voice and my family's voice heard because it is very important that you get the other side of this, another perspective. I am here to let everyone know, especially those who are perpetuating the lie that I am oppressed. That I can speak for myself, that I can walk, that I can talk, I can read, I can swim, we are not all the same.

> Despite what Joe Biden says, I also understand how to operate computers, in fact my children built their own computers. And I also want to let everyone know that we are also very capable of inventing, that Blacks can build, that we can become Supreme Court justices, that we can lead armies, that we can break Olympic world records, that we can become NASA mathematicians, and can become pivotal to sending the first American astronauts into space, that we can also become the President of the United States for two terms. That's what Blacks can do, and we are not oppressed. We can do all of this because we live in an incredible country, America, that offers . . . limitless possibilities for all people whom are willing to dream and work hard. That is why I LOVE this country and that is why I oppose Critical Race Theory and anything that resembles it.

> The single biggest obstacle to success for any person is the limitations they place on themselves. It is also the mental insultment (sic) perpetuated by an infectious political party. I believe in higher education, and I believe that representation matters. Studies show time and time again that higher education equates to higher income. Dumbing down education isn't the answer. And you know what else isn't the answer? Telling Black people that they are inferior by suggesting they are oppressed simply because of their skin color. THAT is discrimination and THAT is racist . . . .. Imagine how awkward . . . that first day would be when a black child walks into a school that is teaching CRT and they don't know if their relationships are authentic or out of pure pity. Imagine how that must feel.[10] . . . I believe that CRT is the new Jim Crow . . . .

In addition to CRT materials, under pressure by CLS-LGTQ thinkers, schools have embraced and, in some cases, actively promoted student questioning and then self-selecting their gender based upon how they feel, using a "gender unicorn" to teach the youngest children, in addition to sexually explicit books discussing sodomy, etc. The American Bar Association argues that such education is a path to better public health because in the past transgender students have been stigmatized and discriminated against (Bittker 2022). The Analytic Philosophy/CLS approach seems to lead to frustrating and sometimes

---

[10] (Here's Why This Idaho Mother Opposes Critical Race Theory 2021). YouTube has other videos of parents' similar objections, as Google's algorithms seem to be hiding them, they are more easily found by searching YouTube itself.

nonsensical conclusions, but there is another tradition that may provide an alternative and more productive solution, enabling truth-finding and hence justice.

What Kelsen, Hart, Rawls, Dworkin, CLS, and CRT all share is a reliance on exploration: a belief that law and every social practice is to be explained by reference to an initially hidden structure. The belief that this is the correct method of analysis relies on an assumption that social phenomena can be (and should be) explained in the same way we explain physical sciences. This is scientism, and positivism is just one type of scientism. Dworkin rejects positivism, but simply substitutes another hidden-structure theory to explain a universal legal system; thus, he is not rejecting scientism.

All of these thinkers engage in an allegedly social scientific analysis of the social world, promising that once we have access to the initially hidden structure, we can engage in a form of social technology and improve society. The aimed-for social technology is often a form of socialism involving the redistribution of wealth, but especially in view of the CLS-CRT-Fluid-Gender movement, the real problem of hidden structure thinking is that it gives free license for anyone to plug in any political agenda as the purported hidden structure. There is no way to choose among the various hidden structure hypotheses (Rawl, Hart, Dworkin, Unger), no way to verify the truth of what was allegedly hidden. This encourages each new theorist to delegitimate the others by crafting a different hidden structure thesis about the motives and agendas of his or her opponents (Capaldi 1995). **The result is an end to civil discourse, seen (tragically) most especially in the United States, as in the "canceling" of scholars who have presented unpopular views or presented research inconsistent with politically correct views.**

## 4. The Common Law Tradition

While Dworkin's influence has been felt widely in both Europe and the United States since the 1980s and CLS is spreading, these are not the traditional understandings that common law has had of itself, nor are they the current understanding habitually displayed among judges, attorneys, and the populace in the Anglosphere—and a number of things that can be learned by studying that tradition. This section details the growth and development of the common law from within the context of its culture.

### 4.1. English History

As with the Civil Law Tradition, the values and assumptions inherent in common law can be traced back hundreds if not a thousand years. In contrast with the civilian tradition, one basic assumption is that governmental powers are and should be limited and are checked by various methods, including by law and by the populace itself. Another is that law is supreme, above government, and it should not be instrumental—in the language developed by philosopher Michael Oakeshott, a government should be a civil association rather than an enterprise association: its proper purpose is to maintain peace, not improve society (Oakeshott [1983] 1999). The common law conception of the rule of law ("and not of men") and hence justice, like the civil law, developed over a long period of time.

Many cultures, such as China's, Russia's, and others still have a top-down pyramidical social structure wherein the lower echelons exist to serve those in the upper echelons, thus discouraging creativity, individuality, and competition (Znawenski 2012). Through the study of linguistic roots, the unique Western focus on creativity and individuality has been traced back to the Indo-European aristocratic egalitarian and warlike culture that spread from the Ukrainian Steppes starting in the fourth millennium to central and western Europe, the Near East, and India (Duchesne 2012). The last such migration was of the "Germanic barbarians" who dislodged the western Roman Empire, conquering the Continent, and then adopted the top-down, centralized, and collective social view and theory-oriented thought prevalent in the pre-existing Roman and Catholic cultures of southern and eastern Europe, thus sublimating the original Indo-European ethos.

In contrast with southern and eastern Europe, however, the Angles and Saxons who conquered England maintained a less-diluted form of the original Germanic ethos, which

rooted its norms in prior practice rather than philosophical or theological theories and first fought to safeguard their personal autonomy, then did so through institutions such as Parliament and the King's Courts, in such documents as the Magna Carta, and finally through a web of contractual relations among a wide variety of groups. Over time, the ethos inherent in the Indo-European culture led to the full expression of individual potential, channeled into a wide variety of economic, scientific, creative, and political pursuits all over Europe, but particularly in England where there was traditionally less focus on collective society and more on individualism.

Roman historian Tacitus contrasted the tyranny of the Roman Empire with the liberty he saw in the Germanic Tribes that ultimately settled in England: while chiefs could decide minor matters themselves, if a matter was important, it would be debated publicly and collectively in a "Husting" or governing assembly of the free people of the community (Nedzel 2020, p. 16). If they approved of a king's proposal, they would brandish their spears; if they disapproved, they would reject it with murmurs, and if they were really unhappy with a king, they would depose him. Law for them was a common possession of the tribe, perpetuated by word of mouth, and so ingrained as to have become regarded as permanent and unchangeable—unless modified in a husting. These habits of the Germanic tribes were gradually abandoned on the continent after Charlemagne created the Holy Roman Empire but remained in England because it was isolated. In short, the Germanic Tribes maintained their original non-Greco-Roman culture only in England.

William of Normandy conquered England in 1066, but instead of making it part of France (which would have relegated him to the status of a Duke), he preserved its identity as a separate kingdom. First, doing so gave him the status of a king instead of a mere duke; second, it produced more income that way (the Anglo-Saxon tax collection system was better developed than its Norman counterpart), and finally, the Anglo-Saxons were more likely to accept his rule. In order to maintain peace and reduce the likelihood of rebellion, William I formally promised to uphold existing Anglo-Saxon laws and customs, though there were significant changes: he introduced a split society with Normans (a dominant minority) imposing some new rules, a new language, and special feudal courts for the French ruling class and ecclesiastical courts in addition to the existing system of courts.

Over time, the two traditions became amalgamated into one, but the English maintained the tradition of periodically requiring that a king agree (sometimes in writing) to certain concessions before being allowed to rule and they continued the tradition of deposing (and usually killing) kings who ruled badly. At least eight English kings succumbed to this "English Disease," as the French described it, between 1087 and 1688, but research discloses few continental kings deposed or murdered during that period.[11] Under the English version of feudalism, even if the king was sometimes regarded as a divinely ordained ruler, he was also expected to follow English law and when he abused his powers, his barons (and others) united and rose up against him. His powers were further circumscribed by the common law that began to develop under Henry II, by three foundational documents that Kings were forced to sign in order to be allowed to rule (the Charter of Liberties, the Magna Carta, and the Petition of Rights), and eventually by Parliament which alone had the power to create new taxes after Edward I created the House of Commons in 1295 (Nedzel 2020, p. 20). On the surface, both England and the Continent eventually did away with absolutist monarchs. However, the continent maintained some form of the notion that the nation was an enterprise association wherein a dominant government's role is to improve society whereas the English have always generally understood themselves to be individuals and the government to be a civil association with no goal other than protecting peace.

Not only were English attitudes towards rulers different from continental attitudes, but so were the attitudes and habits of the English populace. The liberty Tacitus saw continued through the feudal period and developed into individualism. Historian Alan Macfarlane's

---

[11] Nedzel (2020, pp. 20–21): two French kings (killed by Catholic zealots) one Polish king (killed by a family member) and one Dutch 'king' (William the Silent, 1584, killed for his religious toleration).

research found that the peasant society that existed between Norman Conquest and the industrial revolution of the eighteenth century was very different from that of continental serfs. The English peasants were wealthier, independent, and individualistic as a result of a very different economy, social structure, laws, and political system (Macfarlane 1978, p. 175). Continental visitors to England from the middle of the sixteenth century remarked on their wealth, arrogance, lack of subservience, and that English peasants were "impatient of anything like slavery." (Macfarlane 1978, pp. 173–88). As early as the thirteenth century, there were innumerable licenses to sell and transfer land, indicating a considerable market in freehold properties. Single women could make a will, own property, and enter contracts, and they generally left the family home at adulthood and made their own way in the world, something unknown in continental peasant societies (Macfarlane 1978, p. 131).

The scope of English law was much less intrusive than that of the civilian tradition. For example, in contrast with continental European restraints on inheritance (France still has forced heirship), English men and women could and did (in great numbers) make wills and leave their property to family or other persons as they pleased. Thus, as early as the thirteenth century, children had no automatic inheritance rights and could be left penniless. In explanation, Jurist Henry de Bracton (1210–1268) argued "a citizen could scarcely be found who would undertake a great enterprise in his lifetime if, at his death, he was compelled against his will to leave his estate to ignorant and extravagant children and undeserving wives." (Macfarlane 1978, p. 103). In addition to having a narrower scope of the law and valuing entrepreneurship, Bracton's work showed the English assumption that a king's authority was limited and that he was himself inferior to the law—in contrast to Justinian's Digest's underlying assumptions. Bracton stated that "the king must not be under man but under God and under the Law, because the Law makes the King . . . for there is no Rex where will rules rather than Lex." Furthermore, "if the king should be without a bridle, that is without the law, they ought to put a bridle on him." (Bracton 1569). Bracton may have adopted some of the deductive thought underlying the continental *ius commune*, but he kept the English attitude toward individualism and limited government. That same cultural inheritance shows up again, in even stronger form, in William of Ockham's and Hobbes' philosophy and later in jurist Sir Edward Coke's decisions and legislation.

The work of fourteenth century English theologian and philosopher William of Ockham again demonstrates English antipathy to theory and abstractions, the early existence of English individualism, and the common law view that governmental authority is limited. Ockham argued against the Catholic adoption of Greek abstractions, arguing that whatever knowledge we have of the world is of singular individual things and persons, but not of abstractions such as "telos." He rejected Plato and Aristotle's philosophy, reasoning that it was inconsistent with Christian theology because it contradicted the free will given by God. Ockham's philosophy was inconsistent with Thomistic natural law and included the right to consent to rules and rulers, the right to self-preservation, and the right to private property, as well as a right to private conscience. He was ultimately excommunicated by the Catholic Church because of his views (Seidenthorp 2014, pp. 309–13; discussed in Nedzel and Capaldi 2019, pp. 64–68), but they were consistent with English culture and habit.

Thomas Hobbes (1588–1679) picked up on Ockham's thoughts concerning the implicit consent of the governed and individualism. During the sixteenth and seventeenth centuries of Hobbes' life, much of Western Europe was awash in religious wars between Protestants and Catholics. Many know of Hobbes' *Leviathan* (Hobbes [1651] 2002, I, xiii. 9) only the quote that without government (i.e., "in the state of nature") life would be "solitary, poor, nasty, brutish, and short," but in fact it masterfully explained Ockham's theory of the social contract as an unspoken, implicit agreement not to interfere with each other's liberty. Other concepts that Hobbes introduced and which became fundamental to liberal thought include the natural equality of all men and the artificial character of the political order, that all legitimate political power must be grounded in the consent of the people, and the understanding of negative rights—that people should be free to do whatever the law does not explicitly forbid (the inverse of the Ancient Greek view) (Manent [1987] 1994, pp. 20–38;

See also Nedzel (2020, pp. 32–33). Hobbes' philosophical insights concerning the limits of reason, science, and religion similarly show the singularly English outlook that values custom and experience (inductive reasoning) over theory. His view was that deductive logic although valuable is incomplete and philosophy must include not just the rational study of the universal, but also the study of causes, and it must include experience.[12] According to Hobbes, we use the ability to recall the past and derive universal truths from experience: "[Out] of our conceptions of the past, we make a future." (Oakeshott [1946] 2000, quoting *Leviathan*). However, because experience is ever-changing, there is no eternal, unchanging, or universal truth and therefore all knowledge is conditional.

In terms of cultural context consistent with Ockham's and Hobbes' views, both Sir Thomas Smith (writing in 1565) and De Tocqueville (writing about the United States in 1835–1840) (De Tocqueville [1835–1840] 2010, pp. 1089–29) noticed the high level of social mobility in both England and the United States. Smith commented that England was a land filled with men who of their own free will agreed to live together, an association of equals based on contract (implicit agreement, not written contract) instead of a kingdom of subjects ruled by a superior monarch (Seidenthorp 2014, pp. 176–78). Sir John Fortescue, writing the century before, noted that the rural inhabitants of continental Europe (particularly France) lived in great poverty likely because they were taxed heavily and regularly assaulted and beggared by royal troops. In contrast, English peasants were an association of free men, held together by mutual contracts (implicit agreement) and protected by the common law, trial by jury, and the absence of heavy taxation and torture (Seidenthorp 2014, pp. 180–83). Fortescue believed that these differences between England and Continental Europe dated back to the ancient Britons and were a result of a combination of England's natural fertility, its limited monarchy, and the common law. Certainly, his observations were consistent with what Alan Macfarlane concluded four centuries later. Fortescue believed there had been no basic sociological changes in the customs in the preceding thousand years or more—and looking back to Tacitus, Bracton, and Ockham, he was likely right.

### 4.2. The Development of English Legal Institutions and the Rule of Law

From the twelfth to the eighteenth centuries, law and government developed very differently in England because of the periodic and consistent rejection of continental concepts of law and government by entrepreneurial rulers and ruling bodies in addition to philosophers such as Ockham and Hobbes. As discussed above, William the Conqueror agreed to continuing Anglo-Saxon legal customs, though he created separate feudal courts for the nobility. Those of his successors who flouted this tradition and abused their powers often lost both their thrones and their lives, as described earlier. Other successors, however, built on and developed the tradition of limited government, the rule of law, and justice.

### 4.2.1. Henry I, Edward I, and Henry II

William the Conqueror's son, William II (1060–1100), succeeded his father, but caused problems by violating barons' property rights, oppressing the church, and allowing his soldiers to pillage villages. He was killed by an arrow allegedly fired by one of his own men. His brother and successor, Henry I (1068–1135) was allowed to ascend the throne after signing the Charter of Liberties which formally limited his power over both barons and church officials, and which remains one of England's foundational documents. Henry I also promised to establish peace across England (Hollister 2003). Henry I expanded the royal justice system, strengthened local government, and sternly punished those who violated the king's peace, thus earning the soubriquet "the Lion of Justice." (Green 2009, pp. 242–43). His successor, Henry II, was one of England's most formative rulers, the one who founded both the King's Bench and the Court of Common Pleas as well as the common law with its record keeping, adversarial procedure, and jury system. Because of his quarrel with

---

12 After all, Hobbes was Francis Bacon's secretary, and Bacon reformed the scientific method, emphasizing the need to use observation and inductive analysis, in his 1620 work, *Novum Organum*.

his former friend, Archbishop Becket, over separation of church and state property, Henry II also started the English tradition of training attorneys by apprenticeship rather than at universities, rejecting the teaching of the *ius commune* and thus founding English common law. Two of his monumental changes led to the widespread adoption of the jury trial (which had previously been used primarily in Anglo-Saxon courts): (1) royal recognition of the jury as an effective and enforceable fact-finder (a vast improvement over trial by combat or ordeal, commonly used in Europe at the time); and (2) making jury trials available to ordinary people (Plucknett 1956; Landsman 1983). The habit of systematically recording written decisions led to both the Inns of Court wherein apprentices studied them and the habit of judges of researching recorded decisions to determine pre-existing custom and precedent for later decisions (Berman and Reid 1996; Zywicki 2003).

Henry II was an excellent king, but his successor's (John's) foibles led to the Magna Carta, the second of England's three foundational documents. King John had appropriated close family incomes (both his mother's and his sister's), killed (rather than ransoming) hostages, and stole his bride from a French baron, causing a war with Phillip II of France. (He probably would have benefitted from reading the Iliad). He lost both the war and Normandy, and when he demanded that his barons pay additional scutage so that he could challenge Phillip a second time, they refused, forcing him to sign the Magna Carta in 1215 in order to retain his throne. Civil war broke out nevertheless, and John died in 1216 in the fight to retain his throne.

What is generally unknown about the Magna Carta was that it simply restated both pre-existing limits on the crown's power and pre-existing liberties developed by the court system. Among the traditional liberties were the explicit statement that no freeman could be punished except through legal means, the right to obtain a writ of habeas corpus (a demand issued by the Crown) if unlawfully imprisoned, and the right to a jury trial if accused of a crime. The limitations on the king's power included not being allowed to levy taxes without Parliament's approval, not being allowed to delay or refuse justice, and the principle that no one—not even the king—would be allowed to take the law into his own hands (McKechnie 1914). The Magna Carta was confirmed by at least 30 kings after John, each was a solemn assurance that the king would act with regard for the welfare of all subjects and an acknowledgement that the king (as well as his subjects) was subject to the law. While the long progression of kings might not always have kept their promises, the important point is that the English legal system repeatedly reiterated limitations on governmental power.

### 4.2.2. Sir Edward Coke (1552–1634)

Coke was a contemporary of Hobbes, had an amazing career as a barrister, and left several lasting marks on English legal history and its assumption that government should be a civil association with limited powers. Early on, Coke informed James I that he would have to obey the common law. Having been a prominent prosecutor of the Star Chamber under Elizabeth I, Coke was made Chief Justice of the Court of Common Pleas when James I ascended the throne. In 1607, a barrister was arrested by the archbishop, convicted, and imprisoned for violating the king's authority in defending two Puritans accused of violating Anglican Church law by the Star Chamber. When the King's Court reversed that conviction and James I tried to reinstate it, the question was sent to Parliament to determine if the king had the power to withdraw a case from the King's Court. Parliament asked for Coke's opinion, and Coke held that in accord with common law, "the king in his own person cannot adjudge any case," based on prior law and custom, and Parliament agreed. James I was furious. He believed himself to be an absolute monarch. However, having ascended the throne from Scotland, he was new to England and its ways and demanded that Coke explain himself. Coke stated that under English law and custom, the king may decide an issue only where there is no pre-existing common law principle in place. Where law is already in place, the king cannot sit as judge. James I responded that he, as king, had the prerogative of calling judges to account and that he would always protect the

common law. Coke countered that it was the common law that protected the king, not the other way around. James I accused him of treason, but Coke (though he fell flat on the floor and humbly begged the king's forgiveness) quoted Bracton. James I ultimately relented—undoubtedly because Coke had Parliament's backing and James I was aware of the English Disease.

Coke (and Parliament) continued to be a thorn in James I's side. In 1610, James I overspent his income and tried to raise funds by proclaiming new crimes. In the resulting Case on Proclamations, the Privy Council (of which Coke was a member) answered that the king could neither change the common law nor create new crimes by proclamation without Parliament's approval. Still needing money for military expenses, James I asked Parliament for a substantial subsidy. Parliament approved a yearly allowance of 200,000 pounds; in exchange, James grudgingly agreed to give up the power to collect customs duties and other income sources.

Later that same year, in Dr. Bonham's Case, Coke founded judicial review and stated that the common law limited the power of both Parliament and governmental agencies. When Dr. Bonham, a Cambridge graduate, practiced medicine in London without a license, the Royal College of Physicians had him arrested and imprisoned. Coke found for Bonham on false imprisonment because the statute granted the College the power to arrest someone who had committed malpractice, but it did not give the College the power to arrest someone for practicing without a license and had thus exceeded its authority (Stoner 1992, pp. 48–52). As was observed in Parliament in 1610, "the parliament hath his power and authority from the common law, and not the common law from the parliament, And therefore the common law is of more force and strength than the parliament." In other words, Coke found that the College had violated due process and separation of powers requirements by acting as both prosecutor and judge, and that Parliament, like the King, had limited powers and was under the common law.

James I transferred Coke from the Common Pleas court to the King's Bench in 1612, believing that he would cause less trouble there as the purpose of that court was to protect the king's prerogative and possessions. Coke, however, managed to narrow the definition of treason, a remedy English Kings had previously used to rid themselves of difficult subjects. In 1616, however, because Coke mishandled a case involving adultery and poisoning among the nobility and antagonized nearly every lord in the Privy Council, James I had an excuse to permanently bar him from the bench, but that did not stop Coke. Subsequently elected to Parliament in 1621, he lobbied to have the King's power to grant monopolies limited, for which James I sent him to the Tower for several months (but eventually released him). James I died in 1625, to be succeeded by his son, Charles I. Parliament passed Coke's Statute of Monopolies in 1626.

Like his father, James I, Charles I was also raised in Scotland and believed in the divine and absolute right of kings. He repeatedly angered his subjects in both England and Scotland and breached the common law, as a result of which Coke authored and was instrumental in passing the Petition of Right in 1628, to which Charles was forced to assent as a precondition of any future tax grants. Now on a par with the Magna Carta and the Charter of Liberties, the Petition of Right reiterated no taxation without Parliament's consent, no imprisonment without cause, no quartering of soldiers in subjects' homes, and no martial law in peacetime. Sadly, Charles I continued to ignore the common law and anger his subjects by dismissing Parliament, overspending on disastrous military campaigns, and other acts violating English common law, leading to two civil wars. After a trial in which the High Court of Justice declared him guilty of attempting to "uphold in himself an unlimited and tyrannical power to rule according to his will, and to overthrow the rights and liberties of the people," he was executed in 1649 (Gardiner 1906).

### 4.2.3. Scottish Enlightenment: Locke, Montesquieu

England did not see peace after Charles I's execution until 1660 with the Restoration, when Charles II, Charles I's eldest surviving child, was brought back from exile on the

Continent. Unfortunately, Charles II (the Merry Monarch) though publicly popular because of his liveliness and hedonism, made several serious political mistakes by seizing his opponents' estates, replacing judges, packing juries, and supporting religious freedom for Catholics against Parliament's opposition. Locke had written in support of religious tolerance in the face of a vehemently anti-Catholic Parliament and had also opposed absolutist monarchy, along with his benefactor, the Earl of Shaftsbury. To avoid possible prosecution, he fled to France in 1675, returning in 1679, and then again to Holland in 1683, remaining there until William and Mary were crowned in 1688. Locke wrote his two treatises on government in the 1679–1683 interim.

Locke's First Treatise refuted the divine right of kings (understandable given his situation), a concept that later became widely accepted in the Anglosphere. In his Second Treatise, Locke posited that man in the state of nature is created free and equal, and that government is founded on the implied consent of the governed. (As is consistent with long-held English habits and customs). He went on to develop his labor theory of property ownership: while land was originally owned in common, man owns his labor, and once he has used his labor to work the land, he is justified in regarding that land as belonging to him—as long as "there is enough, and as good, left in common for others." (Locke 1740, chp. V, para. 27). Liberty is primary: man is endowed with the liberty to follow his own will in all things unless they are proscribed by law. He describes government's end as the preservation of private property and the peace, safety, and public good of the people. The people are sovereign, not the rulers. Government may not impose or raise taxes without the people's consent, and it serves three functions: executive, legislative, and judicial. Consistent with what Coke had posited, to Locke the government (whether a monarch or anybody with executive power) is subject to law, and if it acts contrary to the trust reposed in it by the people, then the power previously ceded to it returns to the people. Thus, he allows that a revolution may be justified. Consistent with English themes, for Locke the law is supreme, and a tyrannical government can be overthrown.

Sir William Blackstone (1723–1780), whose work was subsequently targeted by Bentham, became known for his Commentaries on the Laws of England, which was used as the authoritative source on common law for almost a century and the primary work studied by lawyers in the nascent United States. The very first chapter of his Commentaries deals with free individuals' absolute rights, while the last chapter discusses the rise, progress, and gradual improvement of the laws of England. His intended audience was future leaders being educated in English universities, and his aim was to replace the university and clerical emphasis on civil and canon law with common law. Going back to Tacitus, he notes the origin of common law in Anglo-Saxon sources predating the Norman Conquest and emphasizes continuity as the source of political liberty. He stressed the extent to which law evolved through the wisdom of generations, and that such evolution is more effective and just than beginning anew: "We inherit an old Gothic castle, erected in the days of chivalry, but fitted up for a modern inhabitant" (Blackstone [1765–1769] 2016), thus building on Hobbes' empiricism.

Montesquieu wrote one of the most important inductive studies of English law by an outsider in Spirit of the Laws (1748, see Montesquieu 1989). A well-read lawyer and landed baron, he spent two years studying England. Like Locke, he does not identify a concept equivalent to the rule of law, but similarly identifies a number of the same mechanisms limiting governmental power: the separation of powers (positing that tyranny results where any two of those three functions are unified in one entity), he approves of the jury, checks and balances in giving the executive a veto over legislation, a legislative override of that veto, and the right of the parties to a lawsuit having the power to object to a partisan judge.

Madison, in Federalist 48, described the Anglo-American concept a bit more accurately: Separation of Powers, as used in the U.S. Constitution, "does not require that the legislative, executive, and judiciary departments, should be wholly unconnected with each other . . . [in fact] unless these departments be so far connected and blended, as to give to each a constitutional control over the others, the degree of separation which the maxim requires, as

essential to a free government, can never in practice be duly maintained." (Madison 1788b). This explanation is very different from the French conception of complete separation of powers. A clear (an amusing) metaphor for the Anglo-American conception is the hand game of rock-paper-scissors, where no one of the three participants (departments) can remain dominant. On a more serious note, it shows that under Anglo-American legal thought, no one governmental entity has the final word, in contrast with the Civilian tradition – the Supreme Court may have the final word concerning the constitutionality of a legislative act, but the Legislature can then moot that decision with a new law.

### 4.3. The American Experience—Founding, Limitations, Checks and Balances, Jury Trial

On the surface, the American Revolution and the French Revolution look very much alike. In both countries, the populace adopted Enlightenment ideals and revolted against a King, the French revolution being somewhat inspired by the American Revolution, which took place approximately 13 years earlier. However, here, appearances are deceiving. The French were fighting for things they had never had (liberty, representative government, freedom from Feudal vestiges, equality, and class mobility), adopted the ideas of the French Enlightenment, and shortly after that suffered widespread violence in the Reign of Terror. The Russian and other revolutions followed a very similar course: a short-lived bourgeois/republican government followed by mass violence and unrest and eventually a totalitarian government that re-established civil order. In contrast, the American Colonists were fighting for what they already had or believed they had. Every American colony was self-governed from its foundation, having its own elected assembly and its own courts in addition to a crown-appointed governor and a crown-approved charter (and those self-governing charters had been endorsed by the Stuart kings with their absolutist pretensions) (Nedzel 2020, pp. 90–92). However, the Americans started to object when the English government started controlling trade and demanding that the Colonists pay taxes to reimburse the Crown for its expenses in defending them during the French and Indian war (1754–1763).

The American Colonies had grown rapidly; by 1740, their population was close to 1 million, 1/6th of England. As the Colonies grew and started producing some marketable commodities (tobacco, timber) and purchasing others (tea, molasses and sugar to make rum), so did England's desire for control, income, and respect—none of which the Colonists were willing to give. As far as they were concerned, they had been governing themselves and defending themselves against Indian attacks for over 100 years. While the original assemblies were not particularly authoritative before 1688, after 1713, they began to display sovereign attributes out of necessity: issuing paper currency, raising armies, setting policy, building infrastructure, setting rules for elections and legislators, and taxing themselves to pay civil servants, including the crown-appointed governors. Friction increased rapidly as England started enforcing taxes and duties on tea and other goods, preventing the importation of sugar from non-English colonies, and declaring local self-governing bodies void after acts of rebellion such as the "Boston Tea Party."

The American character, from its start, was one of self-determination, religiosity, daring, entrepreneurship, and hustle, the characteristics necessary for someone willing to settle in a new, uncharted, and dangerous continent. They were familiar with the common law, and they were incensed when England started enforcing taxes and customs duties because Parliament passed the taxes without giving the Colonists any voice. James Otis, a fiery attorney, popularized the phrase, "Taxation without representation is Tyranny." In 1787, in looking to draft the Declaration of Independence and later the Constitution, they turned to Locke, not Grotius, Pufendorf, or other continental writers. Locke was read widely, in part because he endorsed a right to rebel against a tyrannical government. Other ideas of his were incorporated into U.S. law as well: "were it not for the corruption and viciousness of degenerate Men, there would be no need" for government, whose entire purpose is to punish the evil men in society. Governments only have the power compatible with that end (punishing evil men). They cannot act arbitrarily, depart from established

laws, take anyone's property without his consent, or delegate law-making power to others." Locke's ideas (and other ideas taken from the Scottish Enlightenment) were incorporated into both the Declaration of Independence and the Constitution.

From the beginning, Americans believed that public virtue was an absolute necessity for self-government and that a lack of public virtue led to Rome's fall. It was also widely accepted that direct democracy was inherently unstable, as both Plato and Montesquieu described it because the public is likely to alternate between mob violence and an unreasoning faith in a totalitarian leader. Northern republicans believed, as did John Adams, that they needed to teach their children to value religion, morality, and liberty, and avoid fortune, ease, and elegance to avoid Rome's pitfalls, whilst southern agrarian republicans (e.g., Jefferson and Washington) believed that maintaining public virtue required fiscal independence: owning enough land to provide for oneself and family and having the ability to bear arms to defend oneself. Slavery was the tragic flaw of the United States from its inception[13]—but North and South united in terms of the need for checks and balances to protect against both evil men in government and the inherent tendency for governmental power to increase, something they had seen repeatedly in their decades of self-governance.

As the Declaration of Independence famously stated, "We hold these truths to be self-evident, that all men are created equal, that they are endowed by their Creator with certain unalienable Rights, . . . that to secure these rights Governments are instituted among Men deriving their just powers from the consent of the governed, That whenever any Form of Government becomes destructive of these ends, it is the Right of the People to alter or to abolish it . . . " The Declaration of Independence was described as a promise, the "apple of gold" framed by the silver frame of the Constitution which came later. That Constitution serves the Declaration's promise of "Liberty to all." (Lincoln 1861; see Guelzo 2001). Both documents are foundational to U.S. law.

To limit governmental power, the American Founders relied to some extent on limits they inherited from the common law: the adversarial process, judicial review, and the jury. The adversarial process limits government by demanding transparency in the judging process. Each side presents its case attempting to convince the judge or jury that the other side is less truthful and that the applicable law is in their favor. The jury, if there is one, serves to identify which side is more truthful; it does not pronounce the law—if there is no jury, then the judge decides both fact and law.[14] The judge must ultimately write an opinion explaining his decision, grounding it in established law, which is then verified by at least one reviewing court consisting of (at least) three judges. Judicial review means that courts are charged with reviewing the acts of the legislature and executive regulations to make sure they comply with constitutional mandates—but only in litigated cases where the constitutionality of a statute or regulation is fairly at issue, and private individuals have the power to bring such claims to courts. The jury historically limits both prosecutorial and judicial power in criminal cases (e.g., the O.J. Simpson murder trial). In civil cases, it can check the power of a big-money litigant—the McDonald's hot coffee case is one such example.[15]

The American Founders built additional structural limitations into the U.S. Constitution. To begin with, each branch was given "enumerated" powers—their power was limited to the actions listed in the Constitution, but additionally, each branch could check

---

[13]   However, it must be noted that at the time of the American Founding, $\frac{3}{4}$ of all people on the planet were enslaved, not just Black Americans.

[14]   Contrary to the opinion of many outside the U.S. that this is an anachronism, American lawyers and judges find that given modern technology, lawyers and judges can clearly explain both facts and applicable law in a way most juries can understand, and most of the time jurors make the right decision based on the parties' credibility (Nedzel 2009).

[15]   A fuller explanation of the McDonald's hot coffee case and this function of the jury in U.S. law is set forth in Nedzel (2020, pp. 108–110); see also *The Truth About the McDonald's Coffee Lawsuit* for an amusing (and true) explanation of the ingenious rationale behind Jury's decision. Available online: www.youtube.com/watch?v=Q9DXXCpcz9E.

the others. James Madison (1788c) explained the reasons for such overlapping powers in Federalist 51:

> But the great security against a gradual concentration of the several powers in the same department, consists in giving to those who administer each department the necessary constitutional means and personal motives to resist encroachments of the others . . . . Ambition must be made to counteract ambition. The interest of the man must be connected with the constitutional rights of the place. It may be a reflection on human nature, that such devices should be necessary to control the abuses of government. But what is government itself, but the greatest of all reflections on human nature? If men were angels, no government would be necessary. If angels were to govern men, neither external nor internal controls on government would be necessary. In framing a government which is to be administered by men over men, the great difficulty lies in this: you must first enable the government to control the governed; and in the next place oblige it to control itself.

Other built-in limitations on the power of the U.S. federal government included its compound nature. As Madison indicated in Federalist 45 (Madison 1788a), while a robust federal government was seen as needed to protect against foreign danger and wars among the different states, it would have little power over state governments, which make their own laws concerning all but the small scope of authority delegated to the federal government. Furthermore, the large number of states, with their individual militias, could combine to check an abusive federal government, while the federal government, in combination with some states, could check abusive state governments (as happened during the Civil War and the Civil Rights movement). An electoral college was added to prevent urban, highly-population states from running roughshod over the interests of more rural or agricultural states. In some of the most recent presidential elections won by electoral vote (e.g., Bush v. Gore, Trump v. Clinton), this provision has been questioned—by those on the heavily-populated coasts and urban centers.

*4.4. Dealing with Factions: Federalist 10 and Contrast with France*

During the two years between the drafting of the U.S. Constitution and its ratification in 1789, writers both for and against the proposed Constitution wrote what has now become known as the Federalist Papers and the Anti-Federalist papers. The former compiles one of the best explanations of the Founders' reasoning. Of these, #10 is one of the most powerful, written by James Madison (1787) based on his understanding of David Hume and discussing the problem of factions, i.e., how to handle the diverse public opinions that can tear a democracy apart. Rousseau and his followers argued that factions should be repressed because there could only be one proper, correct, "general will." Madison vehemently disagreed. He acknowledges that factions can be violent and cause instability, injustice, and confusion in a popular government. We might wish that that was not true, but evidence and experience prove otherwise. He defines a faction as a group of citizens, whether a majority or a minority of the whole, who are "united and actuated by some common impulse of passion or of interest, adverse to the rights of other citizens or to the aggregate interests of the community." Another term for factions would be political interest groups. Madison finds only two ways of stopping factions: removing causes or controlling effects.

The first of two ways of removing the causes of faction is to deny citizens the right to disagree, in other words, to destroy "the liberty which is essential" to the existence of factions. Without liberty, there are no factions because liberty is to faction like air is to fire. Madison argues that abolishing liberty to abolish factions is like wanting to annihilate air: while preventing fire, it would also destroy life. The truth in Madison's reasoning can be seen in the consequences of the Jacobins' attempt to abolish factions to establish a "general will"—which led inexorably to mob violence and totalitarian government. Similarly, giving all citizens the same opinions, passions, and interests is just as impracticable as the first is

unwise. As long as man has the liberty to exercise his reason, his reasoning will be fallible, and as long as his reason is connected to his self-love, that will shape his opinions and passions. Furthermore, people's abilities differ, including differences in earning potential, leading to different interests, financial situations, and political opinions. Factions, therefore, are a natural consequence of being human.

Madison continues to consider different kinds of factions and other ways to control them: religious and political differences and attachments to different leaders who compete for power all lead to mutual animosity, and then people aligned with different factions are more inclined to vex and oppress each other than to co-operate for their common good. Sometimes even the most ridiculous distinctions have been enough to lead to violent conflicts, but the most common source of conflict is the unequal distribution of property. Regulating these various interests and preventing any one of them from dominating either judicial decisions or drafting legislation in their favor, says Madison, is the primary task of modern legislation. Unfortunately, we cannot trust that we will have enlightened statesmen who will ensure that one faction recognizes the rights of another. Thus, Madison concludes that we cannot destroy liberty or change man's nature to avoid factions; therefore, we must find a way to control their effects.

Madison then considers ways of doing that. Where a faction with "sinister views" consists of less than a majority, it is not a threat to a regular vote. It may clog the administration or convulse society, but it will be unable to dominate and realize its aims. However, if the sinister faction is in the majority, it will tyrannize minorities. (The example that comes easily to mind is the rise of the Nazi party in Weimar Germany, but there are many others). Thus, that situation must be avoided, and it is the traditional weakness of a pure democracy.

Madison posits that to control the effects of factions in popular government, in a republic (as opposed to a pure democracy), elected representatives serve as a filter to refine and enlarge public views, and those representatives may be less likely to act precipitously and more likely to work for the good of the nation. However, it is always possible that corrupt or ineffective politicians will betray their elector's interests once elected. One must consider the importance of the number of representatives. In this instance, corruption is less likely to predominate where there are more representatives rather than a few. Still, if there are too many representatives, they may not understand their local electors' concerns and will not be able to work together for the public good. Thus, one needs to find the perfect median number of representatives.

Madison described what was generally known at the time, that a small society will have fewer distinct parties and interests and will more frequently form oppressive majorities. In contrast, however, he theorized that with a large and diverse number of parties and interests, it is less likely that a majority of the whole will have a "common motive to invade the rights of other citizens." Even if such a common motive exists, it will be more difficult for them to act in unison to effectuate that common motive. Consequently, he predicted that a large, compound republic stabilizes popular government by balancing factions against each other, which has proven to be the case in the U.S. The U.S. Constitution and the republic it established are the oldest ones in existence, and it is both large and compound: each of the 50 United States has its own constitution and its own separate republican government that is NOT subservient to the federal government. While the U.S. Constitution stipulates that its law and the law of the federal government is supreme, that law applies only in certain, enumerated areas and only where there is a direct conflict between state and federal law ((Nowak and Rotunda 1991, §§9.1–9.3 (*the Preemption Doctrine*)). Contracts, property ownership (both business and land), licensing, as well as family relationships are governed almost entirely by state law, so most of the law an American citizen encounters in everyday life is state law, not federal.

### 4.5. *Nineteenth Century English philosophers: John Stuart Mill (1806–1873) and A.V. Dicey (1835–1922)*

Nineteenth-century legal philosophers further described the British custom of limited government. In *On Liberty*, Mill discussed "the nature and limits of power which can be legitimately exercised by society over the individual," positing that the history of government has been a continuous struggle between liberty and authority (Mill 1989). Governmental power is dangerous. A single ruler can be a tyrant, but the danger posed in a republic or democracy is the tyranny of the majority. Such governments' power must also be limited to protect individuals.

Interestingly, Mill remarked that the strength of the English tradition of limited government does not depend so much on the habit of regarding governmental power with suspicion but instead relies on the British being unaccustomed to being controlled.[16] Mill also remarked that he noticed an increasing inclination worldwide on the part of the government to control individuals through legislation and that this was corrosive on liberty. He stressed that the only proper exercise of power over anyone against his will in a civilized community is to prevent him from harming others.

The term *rule of law* was first popularized by A.V. Dicey, though he may have taken the phrase from the 1610 Petition of Grievances. Like Mill, Dicey was concerned that the obsession with legislated law was destructive of the rule of law, as was what he perceived as the unchecked growth of administrative law in France. Noting that 1914 Constitutional reformers in England were looking for ways to ensure that any law passed by Parliament should be publicly popular (or at least not unpopular) he wrote: "But these schemes make in general little provision for increasing the chance that legislation shall also be wise, . . . ." (Dicey [1915] 1982, p. lxxix). He also argued that Bentham's utilitarianism and insistence that all law be (recently) legislated leads inevitably to socialism and instrumental law and thus is destructive of the rule of law: "The patent opposition between the individualistic liberalism of 1830 and the democratic socialism of 1905 conceals the heavy debt owed by English collectivists to the utilitarian reformer. From Benthamism the socialists of today have inherited a legislative dogma, a legislative instrument, and a legislative tendency . . . ." (Dicey [1917] 2008).

Dicey described three guiding principles that had enabled the stability of the British Empire: (1) Parliament's legislative sovereignty; (2) the supremacy of ordinary law (which he called the rule of law), and (3) the English reliance on written conventions concerning constitutional law (i.e., Magna Carta, etc.) only as a last resort. He traces the history of the concept of the supremacy of the law in England (as was completed above) and then describes three characteristics of the rule of law. First, as James I found to his chagrin, customary law must be supreme and exclude governmental arbitrariness and its broad discretion. Next, there must be equality before the law, meaning that the ordinary law of the land, as administered by ordinary courts, applies equally to the rich, the poor, and governmental officials. Finally, because individual rights in England were grounded in ordinary judicial decisions, they are the source of England's constitution. Where rights are sourced in written constitutions, the danger is that they may be only (as Jefferson also termed it) paper guarantees, with no actual remedy should government intrude on them. The English focused on providing remedies for intrusion on rights rather than declaring something a right. This distinction is now more usually described as negative rights—the right to be free from governmental interference—as opposed to the civilian concept of positive rights described or listed in a constitution.[17]

---

[16] This *cultural* trait was seen again in the twenty-first-century vote for Brexit—the British did not like being subject to control by the European Union and its directives (legislation without representation). As confirmed by Brexit polls, the English voted to leave the EU for two reasons: (1) the E.U. top-down legal system violated the British understanding of the relationship between citizen and government (49%), and (2) EU membership and the requisite acceptance of its directives were seen as a cause of economic problems caused by large numbers of immigrants (approx. 33%) (Ashcroft 2016).

[17] As the author has described elsewhere, the problem with positive rights is that over time they are likely to be interpreted more and more narrowly. See (Nedzel and Block 2007).

*4.6. The 20th Century: Hayek, de Soto, Leoni, and Fuller*

During and after World War 2, Anglospheric jurists were concerned about preventing the perversions of law and popular government (as well as the economic crisis) that had enabled Hitler's dictatorship and atrocities in a country generally as developed and forward-thinking as Germany.

Hayek, trained in both law and economics in Vienna, posited that the rule of law was never so seriously threatened as it had been by legislation-dominated popular government because of the positivist misconception that so long as all actions of the state are duly authorized by legislation, the rule of law will be preserved. "The fact that someone has full legal authority to act in the way he does gives no answer to the question whether the law gives him power to act arbitrarily . . . . It may be that Hitler has obtained his unlimited powers in a strictly constitutional manner, and that whatever he does is therefore legal in the juridical sense. But who would suggest for that reason that the Rule of Law (i.e., justice) still prevails in [Nazi] Germany?" (Hayek [1944] 2007).  Hayek rejected "constructivist rationalism" (scientism) as posited by Bentham, Jhering, Kelsen, and later Hart, Dworkin, and others who assume that social institutions such as legislation and government should be the product of deliberate design. He argued that such approaches are factually false, are connected with a belief in an unlimited "sovereign" power of government, cannot account for unintended consequences of their legislation (and legislators are not held accountable for those consequences), and can lead to a misunderstanding of the very things that make a society great (Hayek [1973] 1983, pp. 5–7).

The positivist/analytic concept of giving a "scientific" account of the law challenges the previous normative framework that informs the law and replaces it with a theoretical and instrumental conception aimed at what some legal philosopher perceives as a desirable human purpose. The instrumental conception of the law invariably and inevitably leads first to socialism and then to totalitarianism because it views law not as consisting of rules that make possible the formation of spontaneous order by the free action of individuals who limit their actions based on those rules, but instead as the instrument by which an individual is made to serve some collective good as determined by the legislative body or the entity that designed the legislation.: "the whole conception of legal positivism . . . is a product of the intentionalist fallacy characteristic of constructivism, a relapse into those design theories of human institutions which stand in irreconcilable conflict with all we know about the evolution of law and most other human institutions."(Hayek [1973] 1983, p. 71). Under these circumstances, one is ruled by experts, not by the law.

As Hayek describes, it is by this means that a positivist-based legal system becomes politicized, as the winning party's "collective good" becomes mandated law over all objections by minority parties. Especially if that process is corrupted by favors granted to special interests, the public's faith in government and the legal system is weakened. Furthermore, as he makes clear in *The Road to Serfdom*, the mere accretion of legislative and regulatory law encourages rent-seeking: It encourages manipulation of such laws, with the rich and better connected able to hire lawyers to help them do so effectively, while the poor, who cannot afford to manipulate the law or governmental agents, cannot comply, forcing them into what economist Hernando de Soto terms "dead capital," a system of unregistered, clandestine small businesses that can neither grow nor defend themselves because they have no legal presence (De Soto [2000] 2007).  All this further leads to the public acceptance of untruths, unfair government actions, corruption, and injustice and its distrust of both law and the government.

Hayek defined the rule of law as meaning that government in all its actions is bound by rules fixed and announced beforehand (Hayek 1955, pp. 33–34). To him, the intelligibility of these norms rests not on a deductive order but instead on customs where an order originally formed itself spontaneously because the individuals followed rules that had not been deliberately made but had risen according to need and were gradually improved upon over time (Ibid., pp. 2–15). It is the observance of common rules that makes the peaceful existence of individuals in society possible, that limit governmental power, and

that order, that rule of law develops spontaneously. (Hayek 1955, pp. 29–34; Hayek [1973] 1983, pp. 35–50, 56–59). Spontaneous order as embedded in practice led to the development of implicit agreement on fundamental principles, which may never have been explicitly expressed, yet which made possible written fundamental laws, i.e., the implied assent theory of government which Fuller elaborated on as well. Hayek further states that the only country that succeeded building the modern conception of liberty under the law from its roots in medieval "liberties" was England. This was partly due to the fact that England escaped a wholesale reception of the late Roman law and with it the conception of law as the creation of some ruler, but it was probably due more to the circumstance that the common law jurists there had developed conceptions somewhat similar to those of the natural law tradition but not couched in the misleading terminology of that school ... The freedom of the British which in the eighteenth century the rest of Europe came so much to admire was thus not, as the British themselves were among the first to believe ... "originally a product of the separation of powers between the legislature and the executive, but rather a result of the fact that the law that governed the decisions of the courts was the common law, a law existing independently of anyone's will and at the same time binding upon and developed by the independent courts; a law with which parliament only rarely interfered and, when it did, mainly only to clear up doubtful points .... " (Hayek [1973] 1983, p. 85).

Bruno Leoni, a 20th Century Italian legal philosopher, similarly found legislated law problematic and preferred the English system, stating that "[c]ontinental European scholars, notwithstanding their wisdom, their learning, and their admiration for the British political system from the times of Montesquieu and Voltaire have not been able to understand the proper meaning of the British Constitution." (Leoni [1961] 1991, p. 59) He argues that the assumption that legislators represent their citizens in the legislative process is utterly inconsistent with the claim that such legislation is based on some scientific or technological process and has led to a kind of schizophrenia in contemporary society.[18] "What happens, in fact, is that a handful of people ... are given the power to decide what everybody must do within vaguely defined limits—if any."

The resulting legislation is a conglomeration of quick and far-reaching remedies against any kind of evil, and what goes unnoticed is that those remedies are often too quick to be efficacious, too unpredictably far-reaching to be beneficial, and too directly connected with the views and interests of a handful of people (the legislators and their friends). The enormous increase in legislation and quasi-legislative (i.e., administrative) activity on the part of governments everywhere has yet to contribute to any certainty of law but has led to unproductive and sometimes nonsensical intrusions on daily life. Leoni does not argue that legislation should be entirely discarded but that it is incompatible with individual initiative and freedom. While the continental legal tradition did not initially gravitate around legislation, it certainly does now—as does the Anglosphere. (Leoni [1963] 1991, p. 12). Moreover, legislation has come to resemble more and more a diktat that winning majorities impose on minorities; the relationship that legislation has to the social opinion of the community in which it operates may be tenuous at best. (Ibid., pp. 17–18). As former Speaker of the House Democrat Nancy Pelosi once infamously said about the 1990 pages of the Obamacare bill when the Republican minority claimed it was so long they did not have time to read it before voting on it: "We have to pass the bill so that you can find out what is in it." (Roff 2010).

Leoni argued that the devotion to legislative supremacy has entitled officials to behave in ways that under previous law would be judged usurpations of power and encroachments on individual freedom, and that concept has also spread to England: traditionally, legislation that was contrary to the common law would be struck down by the courts, but that is no longer the case.[19] In the United States, realist judge Oliver Wendall Holmes Jr.,

---

[18]  Id. at 8–9
[19]  Id. 98–100

who strongly supported legislative supremacy, famously wrote that "I always say, as you know, that if my fellow citizens want to go to Hell, I will help them. It's my job." (Holmes 1920, discussed in Nedzel 2020, pp. 125–27).

Leoni found defining the rule of law challenging because it is a practice, not a theory, distinguishing it from *rechtsstaat*. The latter, he found, because it is a theory focused on legislative law rather than customs developed over time, imports current politics—specifically socialism—into law, giving license to bureaucrats to act arbitrarily. Bureaucrats are self-interested in ensuring the stability of their agencies, increasing their incomes, and increasing the scope of their powers. In contrast, the rule of law, as it developed in England, Leoni argued, leads to liberty, equality under the law, and therefore more trust in and respect for the law. His views likely were the catalyst for Hayek's migration from the *rechtsstaat* approach to his embrace of the common law's conception of the Rule of Law (Zywicki 2015).

Leoni, like Hayek and Fuller, objected to the positivist dichotomy between law and morality: he saw no point in separating them.[20] He also objected to defining law as an obligation and instead saw law as an accumulation of individual claims—much like Fuller, he posits that law (like language) developed spontaneously out of personal interaction and reciprocal claims, an assertion demonstrated in Ellickson's work. He also discusses the differences between a legal philosopher who sees things from a theoretical viewpoint and a legal operator, who deals with the practical results. He posits that the legal philosopher has a more nuanced picture of the law, rather than the black-and-white view of the practicing attorney/advocate (Leoni [1963] 1991, p. 200).

Interestingly, while Fuller (like Leoni) is skeptical of the abstractions produced by legal philosophers, he would probably disagree about the black-and-white view. Common law attorneys, because of adversarial procedure, must always analyze both sides of a client's situation to anticipate the opponent's arguments and either rebut them or encourage their client to settle. We are trained, from the very first classes in law school, to think inductively, to address both sides of an argument, and consequently realize early on that law is rarely ever black and white. Civilian-trained attorneys, for the most part, are not exposed to this type of thinking, nor are they often exposed to inductive thought.[21] As shall be seen, Fuller's method of attacking Hart and Dworkin's theses is empirical: he applies them to hypothetical real-life situations, thus demonstrating that they are not going to be helpful in practice. Research has failed to disclose any instance where either Hart or Dworkin cogently rebutted the reasoning of those examples. Instead, they simply dismissed them.

*4.7. Twentieth Century: Fuller and Oakeshott*

Like Hayek, Lon Fuller was educated in economics and law but studied economics first at Berkeley and then at Stanford before getting his J.D. He taught law at Harvard for many years, beginning before World War 2. During World War 2, he worked at a law firm in labor relations and continued as an arbitrator of labor disputes until 1959, even after resuming his duties at Harvard. His practical experience in negotiation and dispute resolution informed his thinking about the nature of law in that much of it was related to the principle of reciprocity: tacit and implied mutual assent as well as intentional assent (Lacey 2010, p. 10; see also Fuller 1969, p. 23). Thus, his was a modern take on the implied consent/assent theory and the nature of the rule of law: as society develops, people work out patterns of behavior that maintain peace, and those patterns of behavior gradually become the practice of law, no legal theory needed. That process was demonstrated in Robert Ellickson's widely known *Order Without Law: How Neighbors Settle Disputes*, which discussed the fence-in, fence-out rules that developed in the American West when sheep

---

[20] (Leoni [1963] 1991, p. 193). See also (Bertolini 2015, pp. 561–606) (discussing the importance of the concept too Leoni's thought).

[21] (Nedzel 2021). The Author is known internationally for her work teaching civilian-trained attorneys how to translate their thinking to common law "IRAC" analysis and ran Tulane's LL.M. program doing just that for several years.

and cattle farmers' interests collided ([Ellickson 1991](#)). Like Hayek, Fuller was convinced that the positivist distinction between "is" and "ought" fed into Hitler's rise to power, and he explored ways of preventing this from reoccurring.

Through a beautifully drawn (and humorous) parable about a hapless King Rex who wants to be a good ruler, Fuller set forth eight ways an attempt to create and maintain a legal system can fail in *The Morality of Law*. Those eight distinct routes to disaster include (1) failing to set any rules at all so that every issue must be decided on an ad hoc basis; (2) failing to publicize the rules people are expected to observe; (3) enacting legislation retroactively; (4) failing to make rules understandable; (5) enacting contradictory rules; or (6) enacting rules that require conduct that cannot be followed; (7) introducing frequent changes so that those affected cannot orient their actions; and (8) a failing of congruence between rules as announced and as they are administered ([Fuller 1969](#), pp. 33–51). It is for this that Fuller's work is well known, but he goes much further than that to describe what he calls the "inner morality of the law."

Fuller describes two different kinds of morality: the morality of aspiration and the morality of duty. The morality of aspiration is that which the Ancient Greeks described: it is the morality of the fullest realization of human powers, of excellence, of conduct such as befits a human being functioning at his (or her) best. While the morality of aspiration starts at the top of human achievement, the morality of duty starts at the bottom, consisting of the basic rules without which an ordered society is impossible. Thus, the morality of duty is negative in nature, as in the Ten Commandments. In contrast, the morality of aspiration that the Ancient Greeks proposed is positive. Rousseau identified virtue with knowledge and assumed that if men truly understood the good, they would desire it and seek to attain it. Bentham substituted the pleasure principle for the Greeks' excellence—the greatest happiness for the greatest number of people. Fuller posits that Bentham (and Rousseau, and the Ancient Greeks) and those who think like them are unrealistic: there is no way by which the law can compel a man to live up to the excellences of which he is capable; we can only seek to exclude his life from the "grosser and more obvious manifestations of . . . irrationality" and therefore for workable standards, the law must turn to the morality of duty ([Fuller 1969](#), pp. 5–18). Thus, Fuller, like Madison, views humanity as fallible.

In addition to sharing some of Madison's insights on humanity, Fuller shares insights on the nature of freedom and limited government with both Hayek and Oakeshott, and he shares insights on the relationships among common law, spontaneous order, and freedom with Hayek and Leoni. He agrees with Hayek that the classical liberal state promotes meaningful choice and hence freedom because of the shared view that government should provide a common defense, prevent fraud and violence, protect private property, and enforce contracts ([Fuller 1955](#), p. 1322, citing Hayek). Meaningful choice and hence freedom, however, declines where government has an agenda favoring one group over another, for example if it legislates a policy mandating that the production of coal be doubled, that negatively impacts workers' choice of employment ([Fuller 1955](#), p. 1322). As Oakeshott later described it, freedom is maximized in a civil association that does not favor any group over another, in contrast with an enterprise association which limits it. Fuller, like Leoni, found that the market principles and resulting spontaneous order inherent in common law feed into civil association, providing further support for reasoned choice and freedom ([Fuller 1955](#), pp. 1322–24).

Fuller describes three conditions necessary for (moral) duties to arise ([Fuller 1969](#), pp. 23–24). First, he posits that duty develops out of a relationship of reciprocity resulting from a voluntary agreement between parties. Next, the parties' reciprocal performances are, in some sense, equal in value. Finally, the relationships within the society must be sufficiently fluid so that these relationships of duty must be reversible: a duty owed by one person to another today will likely be owed by the second back to the first tomorrow.[22] Thus,

---

[22] *As American Founder Thomas Paine said it*: "Whatever is my right as a man is also the right of another; and it becomes my duty to guarantee as well as to possess." ([Paine 1791](#)).

Fuller agrees with Hayek (whom he cites) that the rule of law is dependent on reciprocity and man's anticipation that a relationship he has today may be reversed tomorrow and that this is a process, not a theory: society must be organized on the market principle, and the rule of law will collapse in any society that abandons it (see Maine 1861). Those duties (primarily negative) that society has identified over time as being the minimal necessary to maintain such fluid reciprocity are what have developed over time into law, which Fuller describes as the "internal morality of law, i.e., the enterprise of subjecting human conduct to the governance of law." (Fuller 1969, pp. 96–97) He carefully distinguishes this from natural law, saying that this has nothing to do with any "brooding omnipresence in the skies," nor does it have any affinity with religious rules such as a bar against contraception. Moral duties are like the "natural laws of carpentry or at least those laws respected by a carpenter who wants the house he builds to remain standing and serve the purpose of those who live in it." Thus, Fuller regards law as a system of minimal duties that developed over time to enable men to live in peace with each other; he was NOT professing natural law. Instead, he argued that law developed gradually as generations wrestled with developing ways of mitigating and minimizing conflict.

English philosopher Michael Oakeshott (1901–1990), like Hart, served during World War 2—he joined after the fall of France in 1940 at the age of 40 and volunteered for the virtually suicidal Special Operations Executive but was turned down because he was "too decidedly British." Educated at Cambridge in political science, he found Representative Democracy the least unsatisfactory form of governance despite its muddle and incoherence because the "imposition of a universal plan of life on a society is at once stupid and immoral." (Oakeshott 1939) After the War, he returned to Cambridge but left it four years later to teach at Oxford. He left Oxford in 1951 for the London School of Economics, where he was appointed Professor of Political Science, remaining there until retiring in 1980.

Oakeshott posited that we cannot even begin to understand the world if we do not first understand ourselves—understanding ourselves is fundamental, and understanding the world is derivative (see O'Sullivan 2003, discussed in Nedzel and Capaldi 2019, p. 244). He posits that there is no such thing as a human telos that aims at some ultimate fulfillment; thus, aiming for the "greatest happiness for the greatest number" is an oxymoron. The predicament we find ourselves in with freedom is that we are continually challenged to create and recreate ourselves and our understanding of the world based on our experience. We employ freedom by using imagination and intelligence but do not exercise those in a vacuum but within an inherited social context. We are free to add to this inheritance or develop it, ignore it, fritter it away, or even reject it. Oakeshott describes the human condition as a conversation within this inheritance: you join the conversation by speaking at first in the voices of others and eventually in your own voice.

Oakeshott posits that rationalism in politics (what I have termed scientism) is the most severe destabilizing threat to modern societies. Rationalists believe that one can stand both inside and outside the universe of discourse and practice, and they reject any analysis that does not terminate in an unassailable timeless abstraction (think Hart, Rawls, Dworkin, Unger, etc.). They reject explication because it can never be final and definitive. A Rationalist, to Oakeshott, is one who values thought free from any obligation to any authority save the authority of reason, seeing himself as the "enemy of authority, of prejudice, of the merely traditional, customary, or habitual." The problem with this stance is its innate arrogance: in bringing his social, political, legal, and institutional inheritance before the tribunal of his intellect, a Rationalist presents an exaggerated view of both his intellectual ability and his opinion of himself. What is to be feared even more than his conceit, argues Oakeshott, is his belief that he is looking for an innocuous power that can be made so great as to control all other powers, his belief that political machinery can take the place of moral and political education, and that there is no knowledge that is not technical knowledge. The Rationalist wants to begin by getting rid of his social inheritance and fill the resulting nothingness with items of knowledge that he abstracts from his personal experience that he believes to be approved by the common "reason" of humankind.

In his essay on the Rule of Law, Oakeshott explains the role of government and rules/laws about how government should be controlled. He posited that the rule of law describes a kind of human relationship, i.e., that between man and government, that, like many relationships, is governed by rules (Oakeshott [1983] 1999). Most such associations have a goal, and he describes them as "enterprise associations": businesses want to earn money, football teams want to win games, etc.

Thinking about the rules/laws of football, to be both just and fair, they cannot favor either side—they are thus non-instrumental. Players have a mutual obligation to play according to the rules and defer to the umpire's decisions. Therefore, there is no rule that Manchester United will always be the top team in England's Premier League. There may be penalties for the non-observance of the rules of the game, but the rules themselves do not presume any recalcitrance on the part of the players; "fair play" means only that one should play the game conscientiously according to the rules (Oakeshott [1983] 1999, pp. 137–38). Interestingly, cricket goes beyond this, demanding that players themselves report when they have broken a rule, even if no one else would or could notice it—a sort of internalization of the rules, a "conscience," if you will.

Moral rules are similarly adverbial. They are not instrumental to the achievement of anything but describe obligations to observe adverbial conditions in performing self-chosen actions. The rule of law therefore stands for a mode of moral association that recognizes the authority of known, non-instrumental rules (laws) that impose obligations to adhere to adverbial conditions in the performance of the self-chosen actions of all who fall within their jurisdiction. They are not promoting a common interest, such as a particular religious view (Oakeshott [1983] 1999, p. 149). In a secular world with a population that is not homogenous, toleration (and thus justice) is best promoted by non-instrumental laws, i.e., laws that do not favor any particular group and are not formulated to accomplish any political goal.[23] Oakeshott describes them as "adverbial"—rules that describe how to do or what not to do, not what to do. So, for example, the rules of the road tell you which side of the road to drive on, but they do not tell you where to go. Criminal laws describe under what conditions killing another human being will be punished (more severely if the killing is planned, less if it is negligent, and perhaps no punishment if it is in self-defense). Thus, government should be a civil association, remaining neutral to any factions and to preserve justice and liberty to the greatest extent possible. There are times when a government must become an enterprise association, e.g., in times of war or perhaps plague—but otherwise, it should be distinct from enterprise associations such as businesses, hospitals, charities, etc., all of whom have goals to pursue, to preserve an a-political, disinterested nature that is likely to promote justice for all.[24]

According to Oakeshott, western societies are made up of autonomous individuals and anti-individuals (Oakeshott 1991). Autonomous individuals internalize societal rules, and their consciences help prevent them from breaking them; they may even make recompense if they find they have inadvertently broken a rule, as in cricket, and are thus those who embrace civil association over enterprise association. They are also the ones who become entrepreneurs, daring to think and act differently from others, daring themselves to create what was not there before, and taking risks believing that society will find their products useful. Anti-individuals are incapable or unwilling to accept personal responsibility for themselves or their actions and are always parasitic on autonomous individuals. They cannot transition from a communal identity to an individual identity, finding the collective identity familiar and comfortable. They think of themselves primarily as a member of a group—hence the origin of "identity politics." Thus, they become resentful of autonomous individuals, even though they want to enjoy the products produced by them. Consequently,

---

23 Prominent American law professor Brian Tamanaha agrees with Oakeshott and Hayek on this point. (Tamanaha 2006).

24 See Oakeshott [1983] (Oakeshott [1983] 1999, p. 155), rejecting abstract rights as fundamental values of the rule of law because they cannot be logically delineated, unlike adverbials: "thou shalt not imprison anyone without due process", etc.

whereas the autonomous individual wants the rule of law so that he can exchange with other autonomous individuals, the anti-individual wants the state to be a new community that provides him with an increasing list of positive rights, economic equality, solidarity, dignity, and etc.

### 4.8. Fuller on Hart, Dworkin, and Analytic Theorists

Fuller objected to Hart's positivistic theory on practical grounds, setting forth his final arguments in the ten years of the debate with Hart in his 1969 revised edition of the *Morality of Law*. Fuller argues that Hart's division of the Rule of Recognition into duties and powers is an untenable distinction because legal rules often implicitly combine the two, and one must look into legislative intent to discern the difference rather than just reading the text of the rule. Additionally, the distinction is impossible to effectuate (Fuller 1969, pp. 134–41). To illustrate this, Fuller uses King Rex to oversimplify. Suppose King Rex's small country has unanimously agreed that the highest legal power rests in Rex, recognizing him as the sole and ultimate source of law, consistent with Hart's "rule of recognition." Hart posits that the rule of recognition is a power-conferring rule. To discourage anarchy, Hart implies that the rule of recognition does not allow the authority conferred to be withdrawn for abuse. Assuming Rex abuses his power by keeping his laws secret from his subjects, and they take his crown away for doing so, it does not matter whether he was deposed because he violated an implied duty or because he exceeded the limits of his power—it is a classic "distinction without a difference."

Next, borrowing a famous example from Wittgenstein, Fuller provides an example of a mother leaving her children with a babysitter, and instructing the babysitter to teach her children a game. The babysitter teaches the children to gamble with dice or play with knives. Does it make sense for the mother first to consider whether the babysitter violated a tacit duty or whether she exceeded her authority before saying truthfully that she did not mean that kind of game and firing the babysitter? Thus, Fuller's objection to Hart's Rule of Recognition is that it does not consider the rule's purpose. An Anglo-American attorney would argue that it is not enough to follow the letter of the law; one must also follow its "spirit."

In a third example, Fuller returns to King Rex. If Rex IV dies and is succeeded by Rex V, then all laws enacted by Rex IV, under the Rule of Recognition, remain unchanged until Rex V changes them. One did not need Hart to explain this; it is a sociological fact described in the eighteenth century by Portalis "*L'expérience prouve que les hommes changent plus facilement de domination que de lois.*" On the other hand, if Brutus, by a *coup d'état* deposes Rex IV in open violation of the accepted rule of succession, Hart's Rule of Recognition would posit that all previous laws will have lost their force, or he could presumably stipulate that by saying nothing, Brutus tacitly re-enacts previous law—but Hart criticized that argument when it was used by Hobbes, Bentham, and Austin. The need for continuity in the law despite changes in government is so apparent that one typically assumes this continuity as a matter of course. It becomes a problem, according to Fuller, only when one attempts to define law as an emanation of formal authority and excludes from its operations the possible influence of human judgment and insight, as positivism tried to do.

Fuller concludes that neither the rule of recognition nor its division into powers versus duties makes any sense because it excludes tacit reciprocity, it does not provide any insight into legal institutions that by their very nature constantly change (such as Parliament), and it is trying to give neat, juristic answers to questions that are essentially issues of sociological fact. Positivism generally sees law as a one-way projection of authority emanating from an authorized source and imposing itself on the citizen, ignoring the tacit cooperation between lawgiver and citizen. It does not ask what law is or does but only from where it came, and positivism does not view the lawgiver as occupying any distinctive office, role, or function. Since the lawgiver is not regarded as having a distinct and limited role, positivists do not consider that any moral code attaches to his performance—but in real life, an ordinary American lawyer or judge is subject to a stringent code of ethics

governing conduct toward clients, fellow lawyers, courts, and the public. It is not a mere restatement of moral principles governing human conduct. The Rules of Professional Conduct (American Bar Association 2020) set forth certain standards that apply to all those in the legal profession. Finally, Fuller objects to the positivist belief that one must separate the purposive effort that goes into making law from the law that emerges from that effort because it would otherwise be impossible to think clearly about the law. He argues that this, and the other tenants of positivism, completely ignore the importance of human interaction, which brings law into existence in the first place, and without which, the law cannot be understood. In other words, Fuller argues that any scientistic theory about law—whether Kelsen's, Rawls', Hart's, or Dworkin's—is necessarily invalid because it begins by taking human interaction out of the equation.

With regard to Dworkin and the analytic movement, Fuller argues that the basic fault of the New Analytical Jurists is the same as the basic fault of utilitarianism—the utilitarian philosophy encourages an intellectually lazy presumption that means are a mere matter of expediency and need not be seriously considered. In a legal system, what is means from one point of view is an end from another, so means and ends are inextricably intertwined. While Dworkin "accepted" Fuller's conclusion that some degree of compliance with Fuller's eight canons of law is necessary to produce or apply any law, he and other analytic legal philosophers ignore the need to consider the purpose of the law. Dworkin, as well as Hart and others, object to Fuller's statement that law has an internal morality, that a legal system, to be considered just and respected, must follow its own impartial processes and apply its laws equally, and that this is the morality of the law. To discredit Fuller, Dworkin and Cohen facetiously argue that there can be an internal morality of even the most disreputable and censurable of human activities, such as when a would-be assassin forgets to load his gun, or a blackmailer is inept. Hart argued that Fuller's position is confusing and nonsensical because one must separate the purpose of law from morality, saying that under this description, even a poisoner's art could have an inner morality.

Fuller initially found this line of argument so bizarre and so perverse as not to deserve an answer, but later recanted that opinion (Fuller 1969, p. 201), stating that his critics' tacit presupposition that the internal morality of law is a mere matter of efficacy propelled him to clarify his position, which he did with an example from Soviet Russia. Apparently in the early 1960s, so many Russians were trading illegally in foreign currencies that the Soviet authorities decided drastic measures were in order, and in May and July of 1961, they passed statutes subjecting such crimes to the death penalty, apparently to convey that they were serious about punishing such economic crimes. When a leading Soviet jurist was asked about why the Soviet Supreme Court was applying that law retroactively in violation of the Soviet 1958 Fundamental Principle of Criminal Procedure, he replied "We lawyers didn't like that." What he was saying was not that it was ineffective, but that it compromised the principle of justice and impaired the integrity of the law. (The bar against retroactive enforcement of new substantive law is one of Fuller's eight canons). Most important, however, to show Fuller's point, the Soviet action impaired the efficacy of law because it undermined public confidence in both law and the legal system. **Fuller's inner morality of the law is a composite of those principles that enable and encourage public confidence in the law and the legal system.**

Dworkin further tried to discredit Fuller's position that the morality inherent in the law includes a principle against contradictory laws: "A legislature adopts a statute with an overlooked inconsistency so fundamental as to make the statute an empty form. Where is the immorality or lapse of moral ideal?" (Fuller 1969, p. 222, citing Dworkin (1965)) Fuller responded that to begin with, Dworkin's example is outlandish, but more important, in such a case and consistent with the previous example, the public's trust in the law is impaired, and so again, the breach of public trust is the immoral act in question.

Fuller further perceives two assumptions underlying his critics' rejection of the concept of the inner morality of the law: (1) a belief in the existence or non-existence of law is, for them, from a moral point of view a matter of indifference; and (2) they assume that

law should be viewed not as the product of purposive interplay between a citizen and his government, but as a one-way projection of authority, originating with government and imposed upon the citizen. What both Dworkin and Hart miss is any recognition of the role legal rules play in making possible an effective realization of morality in human beings' actual behavior. Moral principles do not function in a social vacuum or in anarchy. To live the good life requires more than good intentions, even if they are generally shared; in the modern world, it "requires the support of firm baselines for human interaction, something that only a sound legal system can supply." (Fuller 1969, pp. 204–5) Fuller's underlying assumptions are consistent with the original ethos of English law: that the law itself is a product of prior practice and a stable and common possession of the culture.

For Fuller, governmental respect for the internal morality of law encourages respect for the law and the legal system. Without that faith in and respect for the law and the legal system on behalf of all citizens, whether they are in government or not, society will collapse into anarchy. One of the most critical aspects of the law is how it is interpreted, as Fuller describes it, "the task of maintaining congruence between official action and declared rule" such that interpretation occupies a sensitive, central position in the internal morality of the law, revealing the cooperative nature of the task of maintaining legality. Hart regarded this as a non-issue and that concern about properly interpreting law is a "preoccupation with the penumbra," something that causes only occasional difficulties (Fuller 1969, pp. 224–26). Kelsen similarly dismisses judicial interpretation as simply a form of legislation, the motives which shape legislation by judges being irrelevant for analytical positivism as those that move a legislature to pass one kind of statute instead of another—an issue that belongs to politics and sociology, not juristic analysis. Dworkin makes a similar mistake in equating judicial interpretation with political theorizing. One American Realist even proposed that statutes be treated not as law at all but only as sources of law, ignoring the fact that such statutory law must be applied by bureaucrats, sheriffs, patrolmen, and others who act without judicial guidance—some cooperation concerning methods of statutory interpretation is an absolute necessity. What all of these critics share in dismissing Fuller's position that interpretation is an integral part of the morality of law is their assumption that law must be regarded as a one-way projection of authority, not as a collaborative enterprise. That cooperation is a vital part of the morality of law.

In Fuller's view, law, like language, arises out of human interaction. Suppose we do not agree on the meaning of words. In that case, we cannot communicate with each other, a fundamental principle that reminds one of Mark Twain's great discussion between Huckleberry Finn and his friend Jim, where Huck is trying to simultaneously show off and explain to Jim that he would not understand what a Frenchman is saying. Huck says, "S'pose a man was to come to you and say Polly-voo-franzy, what would you think?" Jim initially responds that he would "bust him over de head," but when Huck explains that the Frenchman was asking if he spoke French, Jim responds, "Well, it's a blame ridicklous way, en I doan' want to hear no mo' 'bout it. Dey ain' no sense in it." (Twain 1885, Chapter 2) Without agreement as to the meaning of words, we cannot communicate. Similarly, without agreement on the meaning of a law, it cannot be a law.

An authoritative and current American definition of justice is "the fair and proper administration of laws" (Garner 2004, p. 881), but there are several different ways the term is used. Popular justice is "demotic justice, usually considered less than fully fair and proper even though it satisfies prevailing public opinion in a particular case"; in contrast, substantial justice is "justice fairly administered according to rules of substantive law, regardless of any procedural errors not affecting the litigant's substantive rights; a fair trial on the merits." Consequently, though CRT, CLS, and Dworkin have influenced legal theorists in the United States as well as some popular movements, the traditional view of justice still survives and proves Fuller's admonition that law, morality, and justice grow out of collaboration and reciprocity.

## 5. Discussion

The long explication of the histories of the two Western legal traditions provides insight into commonalities and differences and spots those concepts with a positive influence on justice and those with a negative influence. Both traditions have remained true to their roots. Still, the current predominance of scientistic, theory-dominated academic views about the nature of law, seen in Hart and Dworkin, is both nonproductive and corrosive on academia itself, on the legal system, on justice, and on the public because it discourages truth-telling in the name of politics. Hidden-structure theories have no limits and have led to the destruction of civil discourse.

The Civilian tradition, from its inception, has valued top-down authority and expertise as well as deductive thought to help bring coherence and predictability to the law. It holds that the government's appropriate role is to improve society. Napoleon's creation of the original French Civil Code was, in many ways very positive. First, it helped stabilize France's juristic recovery by building from law (both *Ius Commune* and the *Coutumes*) that pre-dated the Revolution, it incorporated the liberal values of equality, protection for private property and contract, and it made the law that generally applies to private citizens both coherent and transparent, appropriate for self-government. It eliminated the vestiges of feudalism. The man on the street could easily read it and understand his obligations to family and others. (Having been trained initially in Louisiana's Civilian Tradition and having taught Civil Code topics for many years, the author profoundly values the elegance, organization, consistency, and clarity of French-derived Codes and has a great deal of respect for the intellectual rigor of the Swiss and German-derived Codes). The concept of a Civil Code was also very portable, which is why 90% of all countries have them.

One of the inherent problems with the civilian presumption that the only legitimate law is that which is produced by the general will as embodied in a legislature is that such law embraces the majority rule. Thus, the law is of necessity driven by politics (Talmon [1952] 2021). The common law tradition, with its traditional emphasis on judicial decision and narrow interpretation of legislation, offsets that tendency to some extent, while the complicated legislative process outlined in the U.S. Constitution was intended to discourage all legislation unless a significant consensus was reached by a large number of factions that the legislation is needed and would be efficacious.

The common law tradition, similarly, has maintained much of its essential integrity, despite the same dangers posed by its adoption of legislative supremacy. It values limited government and customary law and balances judicial doctrine against legislative supremacy in self-governing systems. In many cases, the doctrinal law of both systems uses different mechanisms and concepts, but often reaches the same results (see, e.g., Nedzel 1997).

The role of the judiciary at common law and judicial procedure remains very different from that of civil law; nevertheless, the two legal traditions are often amalgamated into something described as "the" Western tradition, as Europe and the United States came to global dominance and both systems share some underlying values. Problems began, however, when the two were conflated by academic theorists who claimed they were applying science to the study of law, looked for underlying hidden structures that they could claim showed legal universality, and also claimed that as morals are "subjective," they belong to politics and are irrelevant to the science of law. This unfortunate habit of claiming the existence of a hidden universal structure began with the positivists and has proceeded through the analytic legal philosophy of Dworkin to the CLS view that all law is illegitimately based on power, that mere power is the "hidden" foundational structure.

The underlying false premise of the scientistic approach is that there is one "true" and "certain" objective and neutral approach from outside human experience. This first premise is patently false: human beings cannot possibly approach the study of a human institution from outside human experience. A second underlying premise corrosive of freedom is that the purpose of law is to shape and direct human life in society. A democratic version of this second premise, grounded originally in Ancient Greece and continuing through Rousseau's view of the law as representing a General Will and the modern world, leads

inexorably to politicization and even totalitarian repression of dissenting views, as those who adopt it believe with religious fervor that theirs is the only "right" way.

The enormous impact of Hart's book further obscured the distinction between the two different conceptions of law. Dworkin rightly criticized Hart for not clarifying the legal system's normative foundation. What Dworkin (and others) failed to do was recognize that there are competing conceptions of the normative foundations of different legal systems. In so doing, curiously, both Hart and Dworkin smuggled in their own political, normative conceptions without actively engaging the prominent alternative conception, i.e., that of the traditional common law, exposited by Dicey/Fuller/Hayek. The irony of this is that both Hart and Dworkin were products of the Anglo-American tradition but, in fact, imposed on it a continental model viewing law as emanating top-down from legal experts rather than a common, cultural possession.

As Fuller pointed out, the scientistic approach excludes the common-sense tacit understanding of the purpose behind the rules. It excludes all implicit reciprocity, all comprehension that law does not develop in a vacuum. Dworkin's approach and approaches like his, which claim that what judges do is not coherent or purposeful, generate distrust in the system, thus giving an opening to those who want to claim that the underlying *grundnorm* is simply power, or racism, or anti-feminism or . . . etc. No legal system is utopic. Still, the scientistic approach necessarily leads to the denigration of the system, enabling majorities to trample on the rights of minorities, and eroding the public's trust in the system and thus potentially doing devastating damage to a stable society, rather than carefully studying how and why those involved in the legal system interact and the strengths and weaknesses of those interactions.

One of the ways Dworkin's work denigrates the law is in his repeated dismissal of the claim that common law judges make decisions fairly and impartially. This is a tragic misconception for someone who clerked for Judge Learned Hand on the United States Second Circuit Court of Appeals, which reviews decisions of federal district courts in New York, Connecticut, and Vermont. Moreover, it shows an almost unbelievable ignorance about how such systems perform. As mentioned earlier, common law practicing attorneys and civil law-trained attorneys think very differently about the law, because of the differences between civilian and adversarial procedure. First, the adversarial process gives each party the incentive to hunt for any untruth or partial truth emitted by the other party; it also encourages them to agree when there is no factual dispute on a matter. Each party has an equal opportunity to present his or her case or rebut the other party's case, and either party can challenge the judge's impartiality (for good reason) on the record. Another judge will review such a challenge to determine if there is a basis for it and if so, the judge will be replaced. Every utterance made and every paper filed in the lower court becomes part of the public record, ensuring transparency. Reviewing lower court decisions to make sure they are fair and impartial as well as consistent with the law has been systematized for many generations. Though formal procedural rule details undoubtedly differ from court to court, the process is generally the same through all common law court systems, as is the way of writing about and analyzing individual legal problems.

American legal reasoning and writing are highly standardized and taught to every law student; that same method is used in every litigant's memorandum of points and authorities submitted to a court and in every court decision. That method is drawn from the common law tradition but has been made somewhat more concise and efficient and is known by the acronym IRAC reasoning (Issue, Rule, Application, Conclusion).[25] Each legal issue becomes the subject of an individual IRAC. The writer begins by identifying the legal concept at issue, then explains the applicable legal authority thoroughly and objectively so that a reader unfamiliar with the concept will have a good grasp of it and be convinced that the writer fully understands it. Every element of the concept must be documented with

---

[25] The author's own textbook on legal reasoning and writing has been used around the world to teach that method to attorneys pursuing American LL.M. degrees: (Nedzel 2021).

appropriate citations. If a statute is at issue, it is presented, and every component of it is explained in light of interpretive (and cited) case law. Once the Rule section is complete, the writer turns to Application, systematically applying the facts of the instant case to every component of the rule and analogizing to the facts of interpretive cases cited in the rule section where appropriate. Finally, the writer states a reasonable conclusion in light of that application. The process is, in fact, an expanded syllogism: the Rule is the major premise, and the Application is the minor premise leading to the conclusion. The difference is that both Rule and Application must be proven by referencing authority and factual analogy or distinction. In contrast, as the author's Chinese LL.M. candidates once explained, in China, the court flatly states the syllogism, with no concern about whether the resulting decision is analogous to previous decisions, so one wonders about the presence of legal coherence and judicial accountability: there is no way to demonstrate that similar cases are decided similarly. The result of the IRAC discipline is that the writer is obligated to prove to the reader that (1) he understands the law and (2) that it applies to the specific facts of the case as he describes.

Having clerked for Judge Carl E. Stewart of the United States Fifth Circuit Court of Appeals[26] (known for his integrity and collegiality) and having communicated with others who clerked for other judges, the author is very familiar with the appellate process, especially in federal courts. The lower court's complete record, its written decision, the appellant's claims about the errors the lower court made, and the appellee's rebuttal of those claims is delivered to the Circuit Court, and three judges and their nine clerks are assigned the task of reviewing the decision. Before an appellate decision is drafted, the entire trial court record is read, and every case, statute, and legal source cited in the lower court's decision is reviewed to make sure it is accurately described, as is every source cited in both parties' memoranda in support of their positions.

One judge (or one of his/her clerks) is then assigned the task of drafting an initial appellate opinion. The three judges then review that opinion in light of the relevant facts and (this is where their experience becomes important) consider whether, as drafted, there is a good "fit" between facts and law. As compared to previous, similar cases, they consider whether the same result should apply or whether the facts of the instant case are distinctive enough that it would be unjust and inconsistent/incoherent for the same result to apply. The appellate opinion may be redrafted several times before a majority of the judges agree that it is a "good fit."

In addition to the detailed process implemented to ensure coherence and impartiality, judges in the U.S. take an oath to follow the law in making their decisions. The vast majority of them take great pride in following the law, whether or not they agree with it. (If they disagree with it, they can write a dissent, which will be included in the public record and in the Reporters that collect and publish judicial opinions).

Mandatory law and professional habits and customs apply in interpreting statutes and rules. One begins with the plain language of the statute, researching prior case law to see if other courts have interpreted that statute's meaning (see Singer 2000; Llewellyn 1950; Posner 1983; Nedzel 2021, pp. 196–206). The researching attorney or deciding judge also considers the purpose of the law, in light of statutory context, and if still unclear, the attorney may look to legislative history to see what the legislature intended in passing the law. A similar process is used in interpreting contracts between private parties.

One must also consider the peer pressure put on trial court judges as well as the codes of conduct that apply to them in measuring objectivity and fairness. While a judge might be tempted to be less than impartial, the fact that his or her opinion will be carefully reviewed and could be overturned means that his reputation would be damaged as a result. Peer pressure is a powerful force encouraging impartiality, as are the applicable judicial codes

---

[26] The Fifth Circuit reviews decisions from federal district courts in Louisiana, Texas, and Mississippi, and is one of the largest of the 13 Circuit Courts of Appeals, as well as having (with the 2nd and the 7th Circuits) one of the strongest reputations.

which dictate that the mere appearance of impropriety is sanctionable.[27] No judge wants to be overturned on appeal. Consequently, though occasionally a judge will be sanctioned, that is a rare occasion and is more likely to be at the local state court level, where judges are elected and thus more likely to be affected by political winds than federal judges who are appointed.

Merely following common law practice from IRAC through the trial and review processes demonstrates the reciprocity which Fuller described. All have been put in place to encourage truth-finding and justice. Most civil law jurisdictions have similarly detailed processes, some more than others (e.g., Switzerland, Germany, Chile). Thus, as Fuller points out, morality is already in the law, its processes, its procedures, its internal social interactions, and its mechanisms to preserve fairness, impartiality, and coherence to the greatest extent possible. We need to examine what is without thinking that we need to develop some abstract theory about what we are doing. Human institutions are not, and can never be, described in the abstract, and pretending we can step outside ourselves to study ourselves is oxymoronic.

## 6. Conclusions

It has been the focus of this article to demonstrate why Dworkin's approach is not only non-productive in the pursuit of truth and justice but even destructive of the legal system (whether common law or civil law). A more productive approach is that demonstrated by Lon Fuller, that of simply examining what is undertaken without abstractions and studying the history of how those habits developed and what purpose they serve, bearing in mind that they are the product of reciprocal human interaction, whether active, tacit, or implied, and the ultimate goal is to encourage and secure positive human interaction.

Will such studies obviate the current lack of truthfulness seen in society? Obviously not, but to the extent that such has been the result of trickle-down of the inherent lack of respect shown in analytic works such as Dworkin's, at least it will not make things worse. To improve truthfulness, we must build the public's trust in the fairness of the legal system and its pursuit of justice, regardless of whether we believe that the purpose of government is to improve society or whether it should be a civil association. (Though the author believes that the Fuller/Oakeshott view is more likely to help depoliticize legal systems and increase public trust). Time and again, it has been shown that legal transplants and programs such as U.S. Aid's Rule of Law Projects do not work because they do not respect the local culture; transplants never develop the same way in a context different from the ones in which they originated, and they are often imposed on unwilling recipients tolerating the impositions primarily because they came with substantial funding—for example, Venezuela's independent judiciary and restructured Supreme Court came at a World Bank investment of USD 35 million, only to be gutted two years later when elected dictator Hugo Chavez rose to power (Garcia-Serra 2001, pp. 263, 276).

In contrast, legal change that comes from within because of consensus that the pre-existing system was lacking, and which incrementally implements an improved institution in competition with the original system is much more likely to be successful. That was how Henry II created the common law courts: the public found his judges were much more likely to be fair and impartial than the 100 courts run by barons, and the jury system was much more likely to determine who was telling the truth than trial by ordeal. Much more recently, Chile's criminal legal system was similarly redone (Nedzel 2010, pp. 102–8 and sources cited therein). A political consensus was reached that the previous system was dysfunctional. Using a substantial amount of Chile's own money, Chilean attorneys designed a new system borrowing concepts from both common law and civilian tradition and putting recent law school graduates into the roles of judge, defense attorney, prosecuting attorney, and victim's advocate. New courts were set up as an alternative that criminal defendants

---

27  Canon 2 Code of Conduct for U.S. Judges (Effective 12 March 2019), available at https://www.uscourts.gov/judges-judgeships/code-conduct-united-states-judges (accessed on 18 January 2023).

could choose and quickly became favored. Within 15 years, the new system replaced the old one and was followed by similar developments in other fields, such as labor and family courts.

Concerning truthfulness, I close with the following quote from Johann Goethe, which seems on point now more than ever before, when we are surrounded by social media giants claiming that they have checked facts when they have not, or who are imposing their political viewpoints on a public that can now find the truth only with great difficulty:

> "Truth has to be repeated constantly, because Error also is being preached all the time, and not just by a few, but by the multitude. In the Press, Encyclopedias, in Schools and Universities, everywhere Error holds sway, feeling happy and comfortable in the knowledge of having Majority on its side."

In this Article, I have tried to tell that Truth.

Nadia E. Nedzel

29 November 2022

**Funding:** This research received no external funding.

**Data Availability Statement:** Not Applicable.

**Acknowledgments:** The Author would like to thank Judge Carl Stewart for his mentorship and philosopher Nicholas Capaldi for his helpful criticism and constant encouragement.

**Conflicts of Interest:** The author declares no conflict of interest.

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
