# Peer review of "Fuller, Dworkin, Scientism, and Liberty: The Dichotomy between Continental and Common Law Traditions and Their Consequences"

_laws, 2022_

Round 1

Reviewer 1 Report

This article is superb. Comments below are purely suggestive and not conditions for acceptance. Several of them are suggestions for shortening the article, which is somewhat long.

Line 739: typo. Article says "distrubution" and I think the author means "distribution"

Lines 762-784: This material is substantively fine. But it feels like a bit of a digression from the central argument of the paper regarding larger jurisprudential questions.

Footnote 105: typo, author means "Berman" (not "Bergman")

lines 1575-1582: Around this paragraph the author may find useful in this discussion this article, which discusses Leoni's crucial influence on Hayek in Hayek's migration from the Rechtstaat approach to the rule of law to embracing the common law: https://papers.ssrn.com/sol3/papers.cfm?abstract_id=2503080. 

Lines 1614-1626: Around this paragraph Fuller's article on "Freedom" might be worth revisiting. Lon L. Fuller, ‘Freedom – A Suggested Analysis’, 68 Harvard Law Review 1305 (1954-1955). It overlaps with Hayek closely.

Paragraph 1627-1648: This paragraph brings to mind Henry Sumner Maine's book discussing the evolution "from status to contract," ie, the development of a social/economic system characterized by reciprocal, consensual obligations rather than feudalism.

Line 1794: Stylistic comment--"positivism posits" sounds sort of funny. Maybe a different word for "posits" there?

Around this paragraph, perhaps Bruno Leoni's idea of "law as claim" might be worth a mention. It is available as an appendix to the revised edition of freedom and the law. Daniele Bertolini has a good article on the importance of that concept to Leoni's thinking, particularly how it reinforces the primacy of private ordering.  https://papers.ssrn.com/sol3/papers.cfm?abstract_id=2543190. I believe Alberto Mingardi has written on that as well.

Part 5 (lines 1896-1945): I wasn't sure what the author was doing with this section. I suspect the idea here was to develop the empirical case for common law versus continental law. But if so, that conclusion isn't developed here. I'm not sure it is necessarily relevant to the overall argument either. I'd suggest the author consider deleting this section.

Part 6 "Discussion": Much of this discussion wasn't obviously necessary to me. I understand the author wants to respond to the critique of the common law process as unfair, but this seems like somewhat of a digression to me (such as discussion of standards of review, etc.). It isn't obvious to me this is necessary to the main jurisprudential and historical arguments of the paper. I'd suggest the author consider dropping most of this or shortening the discussion.

Title: My last comment--I'd consider changing the title. I don't think this article is really about "truth and justice" or about Fuller and Dworkin. The alliteration is clever. But when I read the title, I expected the article to be about Fuller and Dworkin and about "truth and justice." It seems to me that sells the article's importance short in terms of its broad range. It seems to me it is really a comparison of the continental and common law traditions, both historically but also how they play out jurisprudentially and with respect to society generally. Fuller and Dworkin are used as embodiments of those positions, but it is about far more. It sort of reminds me more of Hayek's essay, "Individualism: True and False," which discusses two approaches to liberty (French v. Scottish/British). I'd suggest that is sort of the framing of this piece as well. 

Author Response

Thank you SO much for your detailed and careful review.  I have incorporated the changes you suggested and was delighted to make the acquaintance of articles that are new to me and which further support my analysis of Leoni's, Hayek's, and Fuller's positions.  I eliminated Part 5 as per your suggestion, and did some trimming of the Discussion.  By the way, Todd Zywicki and Alberto Mingardi are both friends of mine, Bertolini's work  is new.  I have tried to get in touch with Alberto as I could not find the work you referenced, but have yet to hear back from him.  If you have further knowledge of the article you mention, please let me know.  Again, thank you for your careful and thoughtful work.

Reviewer 2 Report

Original and arresting thesis.  Persuasiveness can be improved by stronger engagement with recent historical scholarship.

Author Response

Thank you very much!  Please give me a name or two concerning whose recent scholarship you suggest I consider including! -- The article is long as it is, but perhaps I might shorten the discussion of something to add in reference to some of the scholars you have in mind.

Reviewer 3 Report

see attached word document

Author Response

Oh my goodness! Thank you for the very careful reading and kind words!  I have removed Section 5 as per your and another reviewer's suggestion, but feel that trimming the historical section would ultimately weaken my argument by making the same theoretical errors I criticize others of doing, so I will stick to my ponderous, common-law style.  Detailed histories are a way of meeting on common ground, a point of departure for discussion wherein we understand the same concepts. In response to your comments, I have also endeavored to clarify my reasons for discussing CRT, to incorporate it more into my thesis, and I have endeavored to bolster and clarify the discussion section.  With regard to American footnote style, I indicated my source at the beginning of a paragraph, and continued reference generally to the same source throughout the paragraph, in an effort to avoid endless "ibids," and I have been able to add a few more sources.  An important source undergirding my criticism of hidden structure thinking is philosopher Nicholas Capaldi's 1995 article in Argumentation entitled "Scientism, Deconstruction, and Nihilism."